# Erythroid-intrinsic activation of TLR8 impairs erythropoiesis in inherited anemia

Jing Liang [1,2,9], Yang Wan[1,2,3,9], Jie Gao[1,2,9], Lingyue Zheng[1,2,9], Jingwei Wang[1,2], Peng Wu[1,2], Yue Li[1,2], Bingrui Wang[1,2], Ding Wang[1,2], Yige Ma[1,2], Biao Shen[1,2], Xue Lv[1,2], Di Wang[1,2], Na An[4], Xiaoli Ma [5], Guangfeng Geng[4], Jingyuan Tong[1,2], Jinhua Liu[1,2], Guo Chen[4], Meng Gao[6], Ryo Kurita[7], Yukio Nakamura[7], Ping Zhu [1,2], Hang Yin [8], Xiaofan Zhu [1,2,3] ✉ & Lihong Shi [1,2] ✉

Inherited non-hemolytic anemia is a group of rare bone marrow disorders characterized by erythroid defects. Although concerted efforts have been made to explore the underlying pathogenetic mechanisms of these diseases, the understanding of the causative mutations are still incomplete. Here we identify in a diseased pedigree that a gain-of-function mutation in toll-like receptor 8 (*TLR8*) is implicated in inherited non-hemolytic anemia. TLR8 is expressed in erythroid lineage and erythropoiesis is impaired by TLR8 activation whereas enhanced by TLR8 inhibition from erythroid progenitor stage. Mechanistically, TLR8 activation blocks annexin A2 (ANXA2)-mediated plasma membrane localization of STAT5 and disrupts EPO signaling in HuDEP2 cells. TLR8 inhibition improves erythropoiesis in *RPS19*[+/−] HuDEP2 cells and CD34[+] cells from healthy donors and inherited non-hemolytic anemic patients. Collectively, we identify a gene implicated in inherited anemia and a previously undescribed role for TLR8 in erythropoiesis, which could potentially be explored for therapeutic benefit in inherited anemia.

Anemia is the most prevalent hematological abnormality detected in children which could be classified into inherited and acquired anemia[1]. Inherited anemia is further subcategorized into inherited hemolytic anemia and inherited non-hemolytic anemia[1]. Defects of the erythrocytes, such as membrane or hemoglobin (Hb) disorders and metabolic abnormalities, constitute the primary causes of inherited hemolytic anemia, exemplified by conditions such as hereditary spherocytosis, β-thalassemia, and pyruvate kinase deficiency. Whereas inherited non-hemolytic anemia, such as Diamond–Blackfan anemia

(DBA), congenital sideroblastic anemia (CSA), and congenital dyserythropoietic anemia (CDA), results in erythroid hypoproliferation or ineffective erythropoiesis in the bone marrow with multiple etiologies. For example, DBA, resulting from specific mutations in genes encoding ribosome proteins (e.g., *RPS19* and *RPL5*) or involved in ribosomal biosynthesis (e.g., *TSR2*), represents one of the most common forms of inherited non-hemolytic anemia. While CDA, characterized by inefficient erythropoiesis, is caused by mutations in genes including *CDAN1*, *C15orf41*, *SEC23B*, *KIF1*, etc., belonging to a heterogeneous group of

[1]State Key Laboratory of Experimental Hematology, National Clinical Research Center for Blood Diseases, Haihe Laboratory of Cell Ecosystem, Institute of Hematology & Blood Diseases Hospital, Chinese Academy of Medical Sciences & Peking Union Medical College, Tianjin, China. [2]Tianjin Institutes of Health Science, Tianjin 301600, China. [3]Department of pediatric hematology and oncology, Institute of Hematology and Blood Diseases Hospital, Chinese Academy of Medical Sciences & Peking Union Medical College, Tianjin, China. [4]State Key Laboratory of Medicinal Chemical Biology and Frontier of Science Center for Cell Response, College of Life Sciences, Nankai University, Tianjin, China. [5]National Laboratory of Biomacromolecules, Institute of Biophysics, Chinese Academy of Sciences, Beijing, China. [6]Toll Biotech Co. Ltd., Beijing 102200, China. [7]Cell Engineering Division, RIKEN BioResource Center, Tsukuba, Ibaraki, Japan. [8]School of Pharmaceutical Sciences, Beijing Frontier Research Center for Biological Structure, Tsinghua-Peking Center for Life Sciences, Tsinghua University, Beijing 100084, China. [9]These authors contributed equally: Jing Liang, Yang Wan, Jie Gao, Lingyue Zheng. ✉e-mail: xfzhu@ihcams.ac.cn; shilihongxys@ihcams.ac.cn

congenital anemia due to the interference of the erythroid differentiation and proliferation[2–7]. Additionally, mutations in other genes, such as *GATA1, EPO, TPS3*, and *ADA2*, were also reported to be involved in inherited non-hemolytic anemia[8–11]. Nonetheless, the causative genes of partial patients with inherited anemia are still enigmatic. Unraveling the pathogenesis underlying such inherited anemia could provide deeper insights into the underlying mechanisms as well as the potential clinical therapeutics of these diseases.

Erythropoiesis is a tightly coordinated process comprising two primary phases. The initial stage begins with multipotent hematopoietic stem and progenitor cells (HSPCs), which differentiate into erythroid-committed progenitors, termed burst-forming unit-erythroid (BFU-E) and colony-forming unit-erythroid (CFU-E) cells. While the latter phase involves the terminal differentiation of erythroid precursors into enucleated reticulocytes, ultimately maturing into RBCs in the circulation[12]. The cytokine erythropoietin (EPO) is one of the principal factors regulating erythropoiesis[13,14]. The interaction between EPO and EPO receptors (EPOR) induces the dimerization of EPOR and subsequently activates the Janus kinase/signal transducers and activators of transcription (JAK/STAT) signaling pathway[12]. Although EPO signaling pathway has been well-documented[15], novel mediators of EPO signaling pathway have not been thoroughly elucidated. Uncovering new regulators that interfere with EPO signaling is vital for deciphering the control of erythropoiesis and may offer potential therapeutic targets, enhancing the clinical management of anemia in various conditions.

Recently, it has been well-documented that HSPCs express the integral components of immune signaling necessary for the regulation of hematopoiesis[16]. In terms of erythropoiesis, mature RBCs express toll-like receptor 9, which promotes RBC engulfment and clearance by macrophages after activation[17]. Erythroblasts express S100a8 which impairs erythropoiesis in the context of RPS14 haploinsufficiency[18]. It is tempting to speculate that erythroid cells express other immune elements that regulate erythropoiesis.

TLR8, an innate immune receptor sensing single-stranded RNA (ssRNA), is predominantly expressed by cells of myeloid lineage[19,20]. By dissecting a diseased pedigree in which the proband displayed inherited non-hemolytic anemia, we identified a gain-of-function (GOF) mutation in *TLR8* that impaired erythropoiesis. Further studies showed that TLR8 was expressed in erythroid lineage from BFU-E stage and activation of erythroid TLR8 impaired erythropoiesis by disturbing the differentiation, proliferation, and survival of erythroid progenitors. Mechanistic investigation revealed a role for TLR8 in ANXA2-mediated plasma membrane localization and activation of STAT5 which ultimately interfered with EPO signaling. TLR8 inhibition improved erythropoiesis in *RPS19*[+/−] HuDEP2 cells and CD34[+] cells from both healthy donors and inherited non-hemolytic anemic patients. Collectively, our findings report a novel causative gene for inherited anemia and offer novel insights into the pathogenesis of inherited non-hemolytic anemia and the network regulating erythropoiesis. More importantly, our study indicates that TLR8 could be the novel therapeutic target for pathological erythropoiesis.

## Results
### Identification of a novel *TLR8*[V434L] mutation in the pedigree of a proband exhibiting inherited non-hemolytic anemia
Our pediatric center of blood diseases diagnosed a child (the proband) (G5-4, Fig. 1a, red arrow), who manifested anemia with Hb at 59 g/L and reticulocytes at $0.0025 \times 10^{12}$/L in peripheral blood since the 24th day after birth (Fig. 1b and Supplementary Fig. 1a). BM aspirates demonstrated a paucity of erythroid precursors, while numbers of cells from other hematopoietic lineages appeared approximately normal (Fig. 1c). In addition, the proband was diagnosed as West syndrome at 6 months of age because of the repeated spasms. In terms of anemia, the need for RBC transfusions was ameliorated after the

administration of prednisone, and his hemoglobin was maintained at over 110 g/L afterward (Supplementary Fig. 1b). Interestingly, a routine family disease history survey revealed many members had various hematopoietic malignancies and, less frequently, auto-immune diseases (Fig. 1a and Supplementary Table 1). For example, the maternal grandmother (G3-10, still living) had been diagnosed with pure red cell aplastic anemia at the age of 45 (Supplementary Table 1). It implies a potential genetic lesion(s) in this pedigree.

Given the frequency of BM failure within the pedigree, we examined known recurrent mutations of inherited bone marrow failure syndromes, such as mutations in genes encoding ribosomal proteins (RP), *ALAS2, GATA1, CDAN1, EPO, EPOR*, etc. (Supplementary Data 1). Strikingly, we did not detect any known pathogenic mutations. Next, to explore the pathogenic mutation(s) in this pedigree, PBMNCs collected from the proband, his parents, uncle, and grandparents (Fig. 1a, the symbols with red border) were subjected to whole-genome sequencing (Fig. 1d). The results showed that the mother and grandmother shared the most mutations with the proband and most shared mutations were heterozygous, suggesting that the genetic lesion(s) was potentially maternally-derived (Supplementary Fig. 1c, d). Regarding the pedigree, we made certain observations: (i) more male patients exhibited severe and early onset of diseases; (ii) the grandmother suffered from pure red cell aplasia; (iii) the mother did not manifest any discernable hematopoietic abnormalities but had experienced recurrent spontaneous abortions and his father, uncle as well as grandfather were healthy. Therefore, principles screening the causative mutation(s) were determined: (a) the proband harbored a homozygous or hemizygous (if it is X-liked) mutation; (b) grandmother could harbor a heterozygous mutation; (c) father, uncle, and grandfather could harbor heterozygous mutation or no mutation; (d) no restriction to mother (Fig. 1d). With these principles, we identified a G to C transition (c.1300 G > C; p.V434L) in exon 3 of *TLR8* (*TLR8*[V434L]) at chromosome X: 12938459 (GRCh37 assembly) (Fig. 1d). Sanger sequencing confirmed this finding and the proband is hemizygous for this mutation (Fig. 1e).

### *TLR8*[V434L] mutation confers hypersensitivity to agonists and hyperactivation
To understand how *TLR8*[V434L] influences TLR8 functionality, we first performed luciferase reporter assays in human embryonic kidney (HEK) 293T cells using pGL6-TA plasmids containing NF-κB response elements, given that NF-κB is a well-documented downstream target of TLR8. Using two well-established TLR8 agonists, R848 and VTX2337, we found that NF-κB was activated more strongly in *TLR8*[V434L] than in *TLR8*[WT] cells upon agonist stimulation (Fig. 2a). TLR8 expression was comparable between two groups after VTX2337 treatment which excluded the possible interference of differential TLR8 expression to the results (Supplementary Fig. 2a, b). Thus, *TLR8*[V434L] could be a gain-of-functional mutation. To understand the potential mechanism of p.V434L enhancing TLR8 functionality, we carried out structural modeling of the activated TLR8[WT] and TLR8[V434L] with ligand ligation. The ligand binding stimulates the conformational reorganization of an inactive TLR8 dimer to form an active one which possesses greater dimerization interface[21]. V434 is located in the leucine-rich region of the interface (Fig. 2b). Given leucine residue confers greater hydrophobicity, we reasoned that the p.V434L mutation enhanced hydrophobic interactions at the interface of the activated TLR8 dimer. Actually, the distance between two TLR8[V434L] protomers was closer than the TLR8[WT] protomers, suggestive of the easier activation status (Fig. 2b).

To confirm the increased sensitivity of TLR8[V434L] mutation to agonists, we generated *TLR8*[V434L] mutant THP1 cell lines in situ using CRISPR/Cas9 gene-editing technology due to the abundant expression of TLR8 and well-documented signaling pathways mediated by TLR8 in these cells (Fig. 2c). As the expression of TLR8 was comparable among

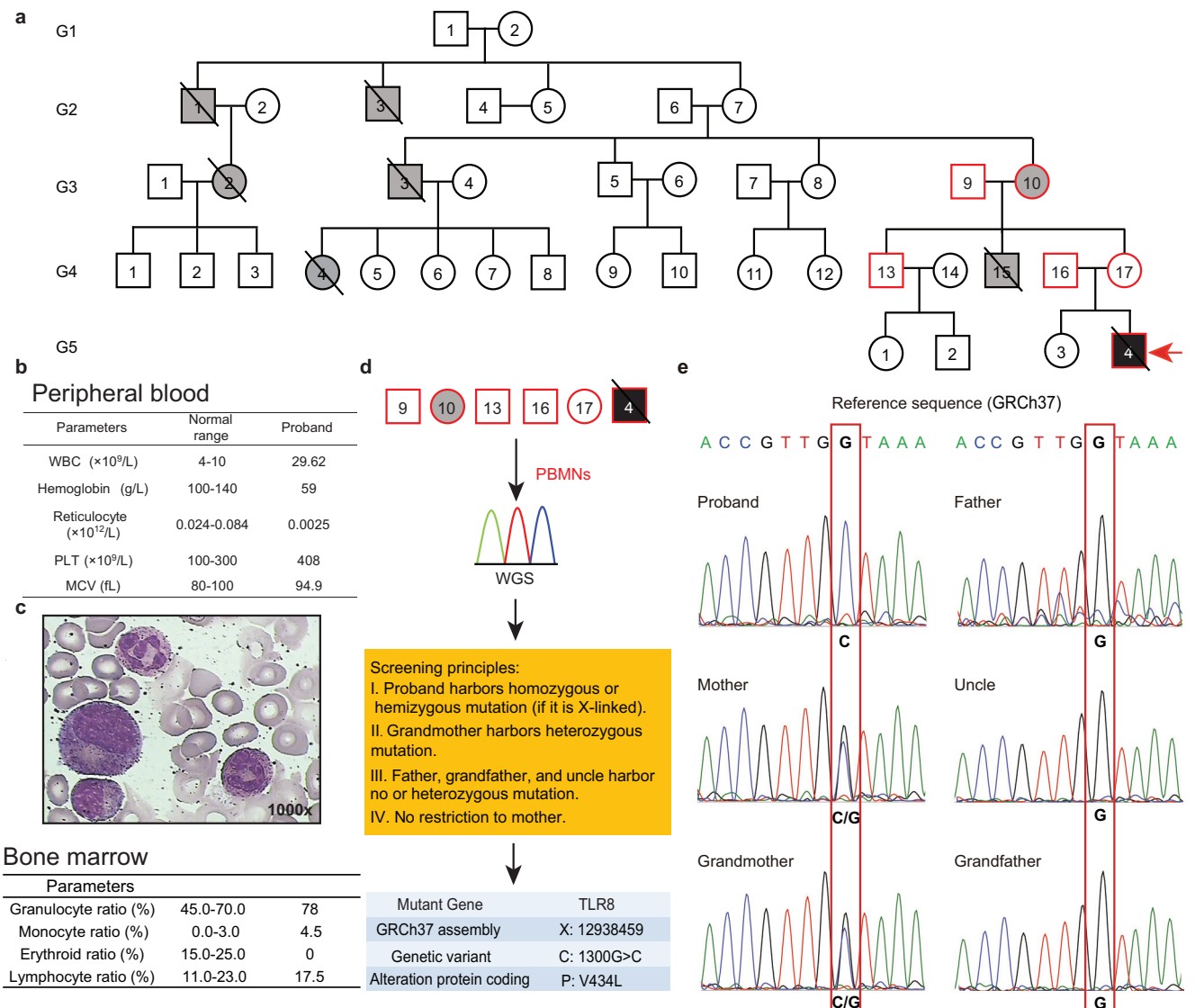

**Fig. 1 | *TLR8^{V434L}* mutation was identified in the pedigree of a proband exhibiting inherited non-hemolytic anemia. a** The pedigree of the proband displaying inherited non-hemolytic anemia. Gray-filled symbols indicate family members suffering from a broad range of diseases, red borders indicate family members whose PBMNCs were harvested for whole-genome sequencing (WGS), red arrow indicates the proband, and symbols with slashes represent deceased family members. **b** Parameters of PB of the proband. **c** Representative Wright-Giemsa staining of bone marrow aspirates and bone marrow cell count of the proband (1000× magnification). **d** Workflow for WGS and mutation screening strategy. "No restriction to mother" indicated no presumption about whether the mother carried pathogenic mutation or not. **e** Reference sequence for WT *TLR8* and corresponding *TLR8* gene sequences of the proband, mother, grandmother, father, grandfather, and uncle (obtained by Sanger sequencing).

isogenic and *TLR8^{V434L}* mutant THP1 cells (Supplementary Fig. 2c, d), the mutant cells produced higher IL-6 than the isogenic CTRL cells after VTX2337 treatment (Fig. 2d). Notably, comparing with isogenic CTRL cells, *TLR8^{V434L}* THP1 cells responded to lower concentration of the agonist, indicating that *TLR8^{V434L}* THP1 cells were conferred with higher sensitivity to TLR8 agonist (Fig. 2d). Consistently, transcriptomics showed that most canonical downstream cytokines regulated by TLR8 were increased in PBMNCs of the proband compared with his mother (Supplementary Fig. 2e). These results were consistent with the structural remodeling.

As the proband exhibited erythroid defects, we sought to uncover the effects of *TLR8^{V434L}* on erythropoiesis. To this end, we constructed *TLR8^{V434L}* HuDEP2 cell lines using CRISPR/Cas9-mediated gene-editing technology. HuDEP2 cells are the immortalized human erythroid progenitors derived from primary cord blood CD34⁺ cells[22]. Interestingly, *TLR8^{V434L}* mutant cells showed less viability and more severely

reduced cell number after VTX2337 treatment compared with the isogenic CTRL cells (Fig. 2e, f). Remarkably, lower concentration of VTX2337 impaired the cell viability of *TLR8^{V434L}* mutant cells other than the isogenic CTRL cells (Supplementary Fig. 2f), indicating *TLR8^{V434L}* mutant erythroid cells were more sensitive to TLR8 agonist than the isogenic CTRL cells.

Besides, we constructed *TLR8^{V434L}* human embryonic stem cell (hESC) lines using CRISPR/Cas9-mediated gene-editing technology (Supplementary Fig. 2g). TLR8 expression was comparable in FACS-sorted CD45⁺ hematopoietic stem and progenitor cells (HSPCs) from *TLR8^{WT}* and *TLR8^{V434L}* hESCs (Supplementary Fig. 2h). Erythroid differentiation was induced in a liquid culture system using these purified CD45⁺ HSPCs (Fig. 2g). The production of CD71⁺CD235a⁺ erythroid cells was reduced at day 8 and larger cellular size were observed by morphological analysis at day 14 of erythroid differentiation in *TLR8^{V434L}* cells (Fig. 2h–j), which were aggravated by VTX2337

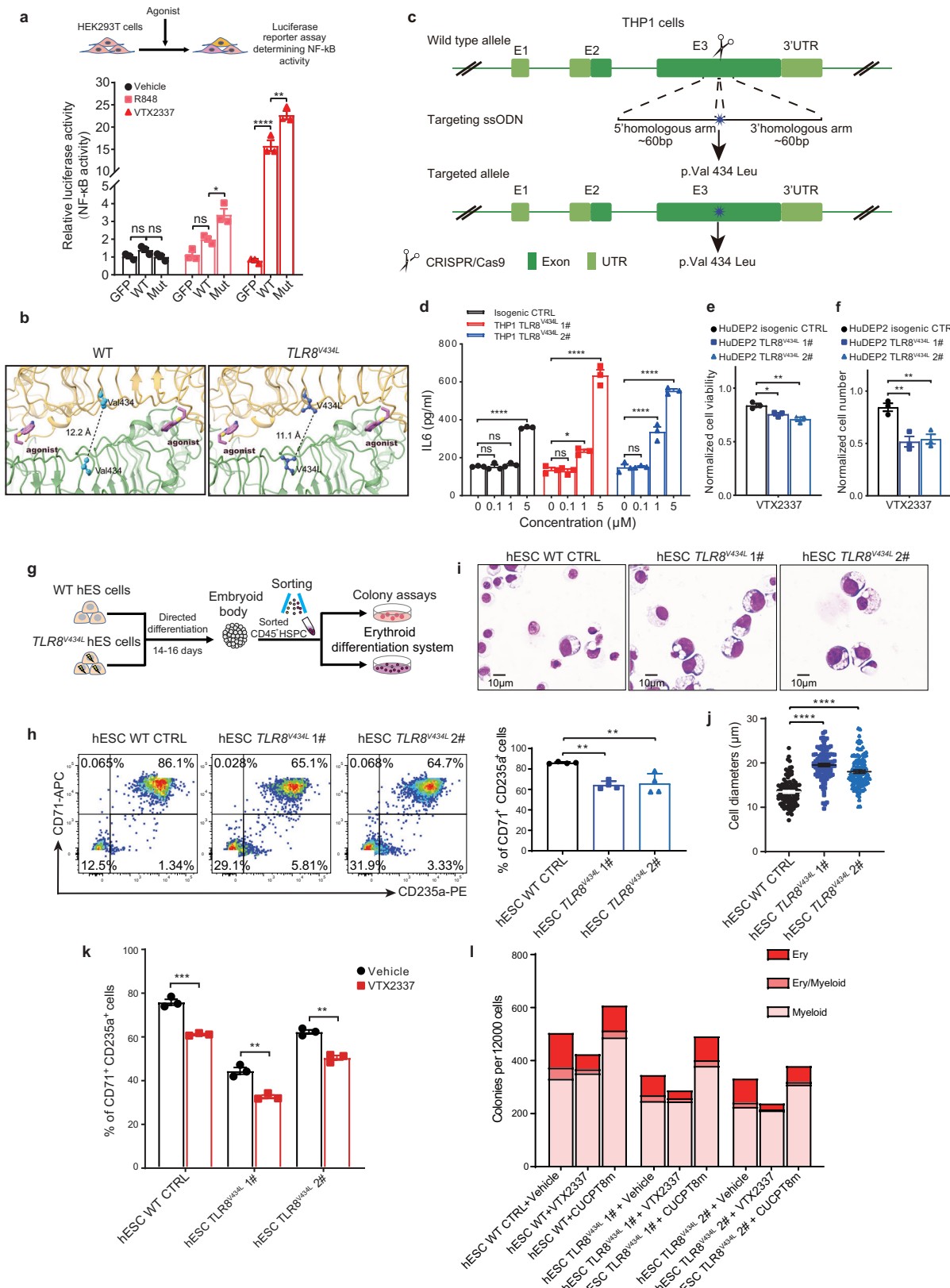

treatment (Fig. 2k). Meanwhile, we observed that VTX2337 treatment greatly induced the total cell apoptosis during erythroid differentiation (Supplementary Fig. 2i). To evaluate whether apoptosis impacts the declined CD71⁺CD235a⁺ in *TLR8^V434L* cells, we examined the apoptosis of CD71⁺CD235a⁺ cells through annexin-V staining and found no evident apoptosis in mutant CD71⁺CD235a⁺ cells with/without

VTX2337 treatment (Supplementary Fig. 2j). Also, the larger cellular size of mutant cells could unlikely arise from the blocked cell proliferation due to comparable Ki-67⁺ cells from each group (Supplementary Fig. 2k). Thus, we speculated that the decreased CD71⁺CD235a⁺ percentage in mutant cells with/without VTX2337 treatment could partially arise from the impaired erythroid

**Fig. 2 | The *TLR8^V434L* missense mutation is gain-of-functional and impairs erythropoiesis. a** Luciferase reporter assays in HEK-293T cells transfected with WT or mutant TLR8 constructs and stimulated with 5 µM R848 or VTX2337. The GFP indicates cells transfected with vector (n = 3 biologically independent experiments). **b** Computational modeling of WT and mutant TLR8 protein structure. **c** Schematic illustrating CRISPR/Cas9-mediated knock-in strategy of mutant THP1 cells. **d** IL-6 production in THP1 cells responding to different dosages of VTX2337. Each THP1 cell line was induced to differentiate using PMA followed by stimulation (n = 3 biologically independent experiments). **e** Cell viability of isogenic CTRL and *TLR8^V434L* HuDEP2 cells after 10 µM VTX2337 treatment for 24 h (n = 3 biologically independent experiments). The viability was normalized to cells without treatment. **f** Cell numbering of isogenic CTRL and *TLR8^V434L* mutant HuDEP2 cells after 10 µM VTX2337 treatment for 24 h (n = 3 biologically independent experiments). The cell number was normalized to cells without treatment. **g** Schematic illustrating experimental design based on hESCs. **h** Representative flow cytometry showing generation of CD71⁺CD235a⁺ erythroid precursors from

WT and mutant hESCs after 8 days of erythroid differentiation. The histogram shows the percentages of CD71⁺CD235a⁺ erythroid precursors (n = 4 biologically independent experiments). **i** Representative Wright-Giemsa staining of cells derived from erythroid differentiation of CD45⁺ HSPCs on day 14. Scale bar = 10 µm. **j** Histogram showing the diameters of cells derived from erythroid differentiation of CD45⁺ HSPCs on day 14 (n = 106 for WT, n = 99 for 1#, and n = 103 for 2# from biological triplicate). **k** The percentage of CD71⁺CD235a⁺ cells from WT and mutant hESCs treated with 10 µM VTX2337 after erythroid differentiation for 8 days (n = 3 biologically independent experiments). **l** The total number of colonies derived from CD45⁺ HSPCs from WT and TLR8 mutant hESCs with various treatment (n = 3 biologically independent experiments). Data are mean ± SEM; statistical significance was determined using the unpaired two-tailed Student's t-test for two-group comparison or one-way ANOVA for multiple comparisons. *$P < 0.05$, **$P < 0.01$, ***$P < 0.001$, ****$P < 0.0001$, ns = not significant. Source data are provided as a Source Data file.

---

differentiation. Consistently, colony-forming assays with purified CD45⁺ WT and mutant HSPCs from hESCs showed that *TLR8^V434L* cells produced fewer erythroid colonies (Fig. 2l and Supplementary Fig. 2l). Altogether, V434L mutation confers TLR8 hypersensitivity to its ligand and impairs erythropoiesis.

## TLR8 activation impedes erythroid differentiation, proliferation, and survival

TLR8 is an innate immune receptor that plays an important role in response to pathogenic ssRNAs and is highly expressed in monocytes/ macrophages, myeloid dendritic cells, and neutrophils[19]. As *TLR8^V434L* compromised erythroid proliferation and differentiation in HuDEP2 cells and hESCs, we further explored the role of TLR8 in primary erythroid cells from the ex vivo erythroid differentiation system using CB-derived CD34⁺ HSPCs[23,24]. During erythroid differentiation, flow cytometry showed that CD71⁺CD235a⁺ cells emerged on day 4 and accounted for >90% by day 14 (Fig. 3a). Morphological analysis was consistent with these observations (Fig. 3b). Notably, immunoblotting revealed TLR8 was expressed in erythroid cells from days 4 to 14 of erythroid differentiation (Fig. 3c and Supplementary Fig. 3a).

To verify the stage initially expressing TLR8 in erythroid cells, we FACS-sorted BFU-E (CD45⁺CD3⁻CD4⁻CD14⁻CD19⁻CD235a⁻CD41a⁻CD123⁻CD36⁻CD34⁺) and CFU-E (CD45⁺CD3⁻CD4⁻CD14⁻CD19⁻CD235a⁻CD41a⁻CD123⁻CD36⁺CD71⁺) cells[25,26] (Fig. 3d). Colony-forming assay confirmed that >90% of the sorted cells were BFU-E and CFU-E, indicative of the high purity of the sorted cells (Supplementary Fig. 3b). Both immunoblotting and flow cytometry confirmed TLR8 expression in erythroid progenitors (Fig. 3e, f and Supplementary Fig. 3c). Immunofluorescence further revealed its expression and endosomal localization in erythroid progenitors as observed in immune cells[19] (Fig. 3g).

We then determined the role of TLR8 in erythropoiesis by treating erythroid cells with TLR8 agonist VTX2337 and antagonist CUCPT8m at various stages of differentiation[27,28] (Fig. 4a). In an initial test of the effects of different concentrations of VTX2337 and CUCPT8m on erythropoiesis, we found VTX2337 remarkably hampered erythroid differentiation and survival while CUCPT8m promoted erythroid differentiation and certified 10 µM as the optimal concentration for treatment (Supplementary Fig. 3d). We also found that consecutive VTX2337 treatment enhanced total cell apoptosis and impaired their proliferation (Fig. 4b and Supplementary Fig. 3e). To further assess the effect of TLR8 activation on erythroid differentiation, after gating the live cells, we examined the generation of CD71⁺CD235a⁺ erythroid cells after VTX2337 treatment. We revealed that VTX2337 treatment indeed reduced the yield of CD71⁺CD235a⁺ erythroid cells in live cells (Fig. 4c). Besides, the size and number of BFU-E colonies incubated with VTX2337 were greatly decreased (Supplementary Fig. 3f, g). These results suggest that TLR8 activation impedes erythropoiesis.

Conversely, CUCPT8m treatment had little effect on apoptosis whereas promoted erythroid proliferation and differentiation (Fig. 4b, c and Supplementary Fig. 3e). In agreement, TLR8 inhibition with CUCPT8m did not alter the size of BFU-E colonies although increased BFU-E production (Supplementary Fig. 3f, g). Consistently, immunoblotting showed that TLR8 was higher after VTX2337 treatment while showed little change after CUCPT8m treatment during agonist/antagonist challenge (Supplementary Fig. 3h).

To determine the stage-specific effects of TLR8 upon erythropoiesis, erythroid cells were treated with VTX2337 and CUCPT8m in stage-specific manner (Fig. 4a, d, e). Namely, erythroid cells were treated with VTX2337 or CUCPT8m from days 0–4, 4–8, 8–11 and 11–14 and analyzed on days 4, 8, 11, and 14, respectively (Fig. 4a). In line with prior observation, TLR8 activation induced total cell apoptosis between days 4–8, 8–11 and 11–14 (Fig. 4d). Further, we analyzed the generation of CD71⁺CD235a⁺ erythroid cells within live cells and found it was decreased after VTX2337 treatment (Fig. 4e). By contrast, TLR8 inhibition exerted negligible effects on apoptosis whereas promoted differentiation at early differentiation stage (Fig. 4e).

To certify the stage initially disturbed by TLR8, we collected VTX2337- and CUCPT8m-treated cells at days 4 and 8 of differentiation for colony-forming assays. It showed that VTX2337 treatment decreased CFU-E whereas had negligible effect on BFU-E (Fig. 4f, g). Apoptotic analysis indicated enhanced apoptosis of CFU-E cells upon VTX2337 treatment (Supplementary Fig. 3i). Then, we postulated that the defects of erythropoiesis by TLR8 activation induced with VTX2337 treatment was attributed both to the apoptosis of erythroid cells (Fig. 4b, d and Supplementary Fig. 3i) and the impaired erythroid differentiation (Fig. 2h–j).

As VTX2337 inhibited while CUCPT8m promoted erythropoiesis, we investigated the potential rescue effects of CUCPT8m on VTX2337-treated erythroid cells. To this end, the erythroid cells derived from CB-CD34⁺ cells were treated with VTX2337 and CUCPT8m simultaneously from days 0 to 11. We found the erythropoiesis hampered by VTX2337 was largely rescued by CUCPT8m treatment (Fig. 4h–j). Therefore, TLR8 activation impairs erythropoiesis from progenitor stage. Overall, our results indicate that TLR8 is expressed in erythroid cells and impairs erythroid survival, differentiation and proliferation after activation.

## TLR8 regulates erythropoiesis in erythroid cell-intrinsic manner

TLR8 is predominantly expressed by immune cells whose activation induces cytokine production that potentially suppresses erythropoiesis. Given erythroid cells are intermixed with non-erythroid cells including myeloid progenitors in our erythroid differentiation system, we sought to determine whether TLR8 could affect erythropoiesis intrinsically. To this end, we conducted the inducer-incubator assay (Fig. 5a). We could discriminate the effects of TLR8 on erythropoiesis

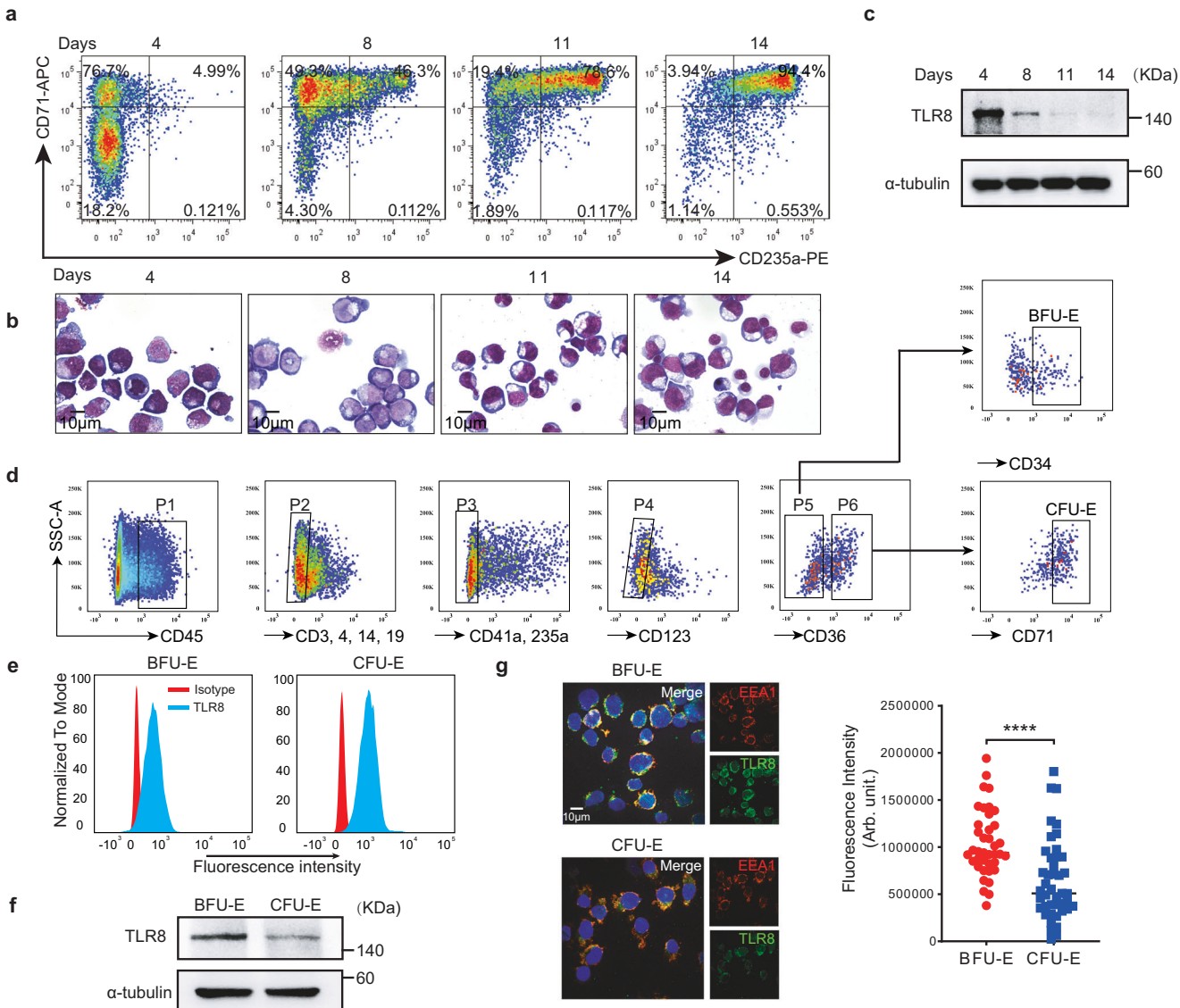

**Fig. 3 | TLR8 is expressed by erythroid cells at early stage. a** Representative flow cytometry and **b** Wright-Giemsa staining of erythroid precursors of cord blood (CB)-derived CD34[+] HSPCs. **c** Immunoblotting of TLR8 in erythroid cells at the indicated time points (n = 3 biologically independent experiments). **d** Gating strategy isolating BFU-E and CFU-E cells. **e** Flow cytometry of TLR8 in BFU-E and CFU-E cells (n = 3 biologically independent experiments). **f** Immunoblotting of

TLR8 in erythroid progenitors (n = 3 biologically independent experiments). **g** Immunofluorescence of TLR8 in BFU-E and CFU-E cells (EEA1, early endosome antigen 1) and the quantitation (n = 39 for BFU-E, n = 43 for CFU-E from biological triplicate); scale bar = 10 μm. Data are mean ± SEM; statistical significance was determined using the unpaired two-tailed Student's t-test for two groups comparison. ****$P < 0.0001$. Source data are provided as a Source Data file.

by solely observing incubator cells that were cultured with conditioned medium (CM) deriving from inducer cells pre-incubated with VTX2337/CUCPT8m. Incubator cells on days 4 and 8 of erythroid differentiation were subjected to colony-forming assays. We found no significant difference in either BFU-E or CFU-E from incubator cells (Fig. 5b, c). Additionally, the percentage of CD71[+]CD235a[+] cells, cell cycle distribution, and apoptosis of incubator cells were comparable among all groups on day 11 of differentiation (Fig. 5d–f). Moreover, we assessed the production of cytokines regulated by TLR8 in inducer cells on days 4 and 8 of erythroid differentiation upon VTX2337 or CUCPT8m treatment. The production of the majority of cytokines in CM was not or marginally altered (pg/ml) (Supplementary Fig. 4a). Thus, the affected erythropoiesis in VTX2337/CUCPT8m-treated cells was unlikely due to the indirect effects exerted by non-erythroid cells after TLR8 manipulation.

To further corroborate TLR8 regulated erythropoiesis in intrinsic manner, we purified CFU-E cells by FACS and performed colony-

forming assays (Fig. 5g). It was shown that CFU-E production was remarkably reduced by VTX2337 treatment whereas significantly improved by CUCPT8m treatment (Fig. 5h, i). More importantly, single FACS-sorted CFU-E cell was induced to differentiate accompanied with VTX2337 or CUCPT8m treatment (Fig. 5j). It showed that cell proliferation was greatly impaired by VTX2337 whereas improved by CUCPT8m treatment (Fig. 5k). Besides, erythroid differentiation (CD71[+]CD235a[+] cells) was improved by CUCPT8m treatment (Fig. 5l, m). Altogether, these results indicate that TLR8 regulates erythropoiesis in erythroid cell-intrinsic manner.

### TLR8 activation disrupts EPO signaling by attenuating ANXA2-mediated plasma membrane localization and activation of STAT5

We then aimed to unravel the underlying mechanisms of erythropoiesis regulation by TLR8. As TLR8 activation induced apoptosis in erythroid cells and impaired their differentiation and proliferation which

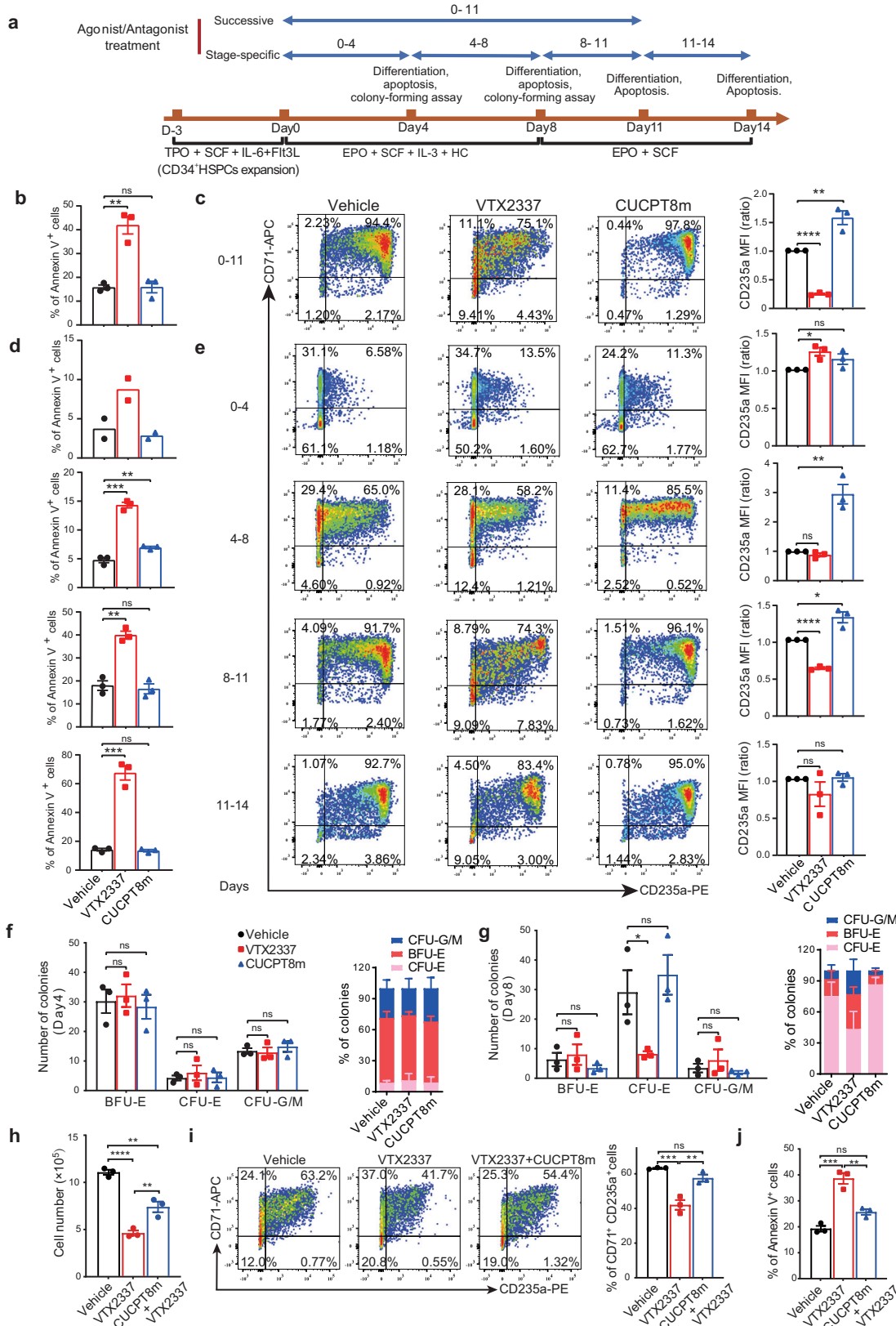

phenocopied the disruption to EPO signaling, we first investigated the potential link between EPO signaling and TLR8 signaling. To this end, we prepared the differentiation medium as vehicle only, EPO only, CUCPT8m only, VTX2337 only, a combination of EPO and CUCPT8m, or a combination of EPO and VTX2337. The differentiation of erythroid cells after each treatment from days 0 to 8 of erythroid differentiation was analyzed. We found that the combined treatment of EPO and CUCPT8m further promoted the production of CD71⁺CD235a⁺ cells and cell proliferation compared with EPO-treated cells whereas CUCPT8m treatment without EPO showed no effects (Supplementary Fig. 4b, c). Conversely, erythropoiesis-promoting effects of EPO were largely impaired by VTX2337 (Supplementary Fig. 4b, c). These results

**Fig. 4 | TLR8 activation impairs erythropoiesis from erythroid progenitor stage. a** Schematic illustrating experimental design. **b** Percentage of annexin-V$^+$ cells after treatment of VTX2337 or CUCPT8m for 11 days (n = 3 biologically independent experiments). **c** Representative flow cytometry indicating the effects of consecutive treatment on the generation of CD71$^+$CD235a$^+$ cells on day 11 (n = 3 biologically independent experiments) and the statistics of CD235a expression. **d** Histograms showing the percentage of annexin-V$^+$ cells on days 4, 8, 11, and 14 (n = 3 biologically independent experiments except for day 4 (n = 2)). **e** Flow cytometry showing the effects of treatments on CD71$^+$CD235a$^+$ cells generation on days 4, 8, 11, and 14 (n = 3 biologically independent experiments) and the statistics

of CD235a expression. The numbers and the percentages of colonies generated by cells treated with VTX2337 or CUCPT8m from days 0 to 4 (**f**) or days 0–8 (**g**) of erythroid differentiation (n = 3 biologically independent experiments). **h** Cell number, **i** erythroid differentiation, and **j** apoptosis of cells treated with 5 μM VTX2337, 10 μM CUCPT8m, or both from days 0 to 11 during erythroid differentiation (n = 3 biologically independent experiments). Data are mean ± SEM; statistical significance was determined using the unpaired two-tailed Student's t-test for two groups comparison or one-way ANOVA for multiple comparisons. *$P < 0.05$, **$P < 0.01$, ***$P < 0.001$, ****$P < 0.0001$, ns = not significant. Source data are provided as a Source Data file.

were confirmed when erythroid cells on day 4 of differentiation were treated during days 4–8 of erythroid differentiation which were primarily consisted of erythroid progenitor cells (Fig. 6a). These results suggest the effects of TLR8 on erythropoiesis were associated with EPO signaling.

To further validate such hypothesis, activation of STAT5, the key downstream effector of EPO signaling in erythroid cells, was determined. We first assessed the level of phosphorylated Y694-STAT5 (p-STAT5) with increasing EPO dosages with/without VTX2337 treatment in HuDEP2 cells. The results showed that TLR8 activation significantly decreased STAT5 phosphorylation after EPO challenge for 2 h (Supplementary Fig. 4d, e). Then, we detected VTX2337 treatment decreased p-STAT5 in primary erythroid cells on day 11 of differentiation while CUCPT8m treatment increased p-STAT5 (Fig. 6b). Notably, neither VTX2337 nor CUCPT8m affected GATA1 expression (Fig. 6b). Consistently, *BCL-xL*, the direct target of STAT5, was decreased by VTX2337 while increased by CUCPT8m treatment (Supplementary Fig. 4f). It suggested that *BCL-xL* downregulation potentially contributed to the apoptosis of erythroid cells after TLR8 activation. The dosage- and time-dependency of STAT5 phosphorylation on TLR8 activation were further assessed in HuDEP2 cells. It showed that VTX2337 treatment decreased STAT5 phosphorylation in dosage-dependent manner and significantly decreased STAT5 phosphorylation after treatment for 2 h (Fig. 6c, d).

Next, we interrogated the mechanism by which erythroid TLR8 regulated STAT5 phosphorylation. Unexpectedly, activation of JAK2 and SRC, the upstream kinases phosphorylating STAT5, did not change in primary erythroid cells after VTX2337 and CUCPT8m treatment despite the significant alteration of p-STAT5 (Fig. 6e, f). These results were also observed in HuDEP2 cells (Supplementary Fig. 4g). Increased STAT5 dephosphorylation by phosphatases after TLR8 activation was thus speculated. To test the hypothesis, we performed co-immunoprecipitation of STAT5 in tandem with mass spectrometry in HuDEP2 cells (Supplementary Fig. 4h). Among STAT5 interacting proteins, PGAM5, the serine/threonine/histidine phosphatase, was the only phosphatase pulled down. Surprisingly, phosphatase assays showed PGAM5 failed to dephosphorylate p-STAT5 (Supplementary Fig. 4i). Therefore, decreased STAT5 phosphorylation after TLR8 activation was not due to the enhanced phosphatase activity. Of note, ANXA2 was identified by mass spectrometry with top unique peptide number and abundance (Supplementary Data 2). ANXA2, which plays the crucial role in the protein localization at the plasma membrane, appeared to be a highly promising candidate given the importance of plasma membrane localization of STAT5 for its activation[29–31]. We set out by determining the physical interaction between STAT5 and ANXA2. Co-immunoprecipitation was performed and the physical interaction between ANXA2 and STAT5 was detected. Importantly, the interaction between STAT5 and ANXA2 was attenuated by TLR8 activation (Fig. 6g and Supplementary Fig. 4j). As ANXA2 aids in the protein localization at the plasma membrane, we speculated the decreased p-STAT5 after TLR8 activation was resulted by the defective STAT5 localization at plasma membrane. Indeed, STAT5 diffusively

distributed within cells after TLR8 activation which was accumulated beneath the plasma membrane in vehicle-treated cells (Fig. 6h). It indicated TLR8 activation interrupted the plasma membrane localization of STAT5. These results were further supported by the significantly decreased p-STAT5/STAT5 level in the fractionation of plasma membrane from cells treated with VTX2337 (Supplementary Fig. 4k). To assess the necessity of ANXA2 for STAT5 phosphorylation, ANXA2 was depleted in HuDEP2 cells using specific lentivirus-expressed shRNAs. Flow cytometric analysis showed that the percentage of CD71$^+$CD235a$^+$ cells was greatly reduced after ANXA2 depletion (Fig. 6i). STAT5 phosphorylation also decreased consistently after ANXA2 reduction (Fig. 6j). More importantly, the capacity of CUCPT8m increasing p-STAT5 was limited after ANXA2 reduction (Fig. 6k and Supplementary Fig. 4l). These results indicate that TLR8 regulates STAT5 phosphorylation at least partially through ANXA2.

Furthermore, we deleted *TLR8* in HuDEP2 cells using CRISPR/Cas9-mediated gene-editing technology (Supplementary Fig. 4m). The results showed that TLR8 depletion abolished the increased p-STAT5 by CUCPT8m treatment (Fig. 6l and Supplementary Fig. 4n). Besides, the enhanced cell viability arising from CUCPT8m treatment was diminished in TLR8-depleted cells (Fig. 6m). Collectively, TLR8 impairs erythropoiesis by interfering with ANXA2-mediated plasma membrane localization and the consequent phosphorylation of STAT5.

## TLR8 inhibition promotes erythropoiesis in physiological and pathological erythropoiesis

As CUCPT8m promoted erythropoiesis of CB-derived CD34$^+$ cells, we further examined the effect of CUCPT8m during adult erythropoiesis. To this end, we induced erythroid differentiation of BM-derived CD34$^+$ cells from healthy donors. Flow cytometric analysis conducted on day 11 showed that CUCPT8m promoted erythroid differentiation and increased cell number (Supplementary Fig. 5a, b).

As the proband was diagnosed with inherited non-hemolytic anemia and TLR8 inhibition improved erythropoiesis, we then determined whether TLR8 inhibition with CUCPT8m would have the beneficial impact on inherited anemia. Here, we tested the effect of TLR8 inhibitors in DBA, a disease model of inherited hypoplastic anemia. To this end, we constructed *RPS19* heterozygous knockout (*RPS19*$^{+/-}$) HuDEP2 cells which mimicked the erythroid defects of DBA[26] (Fig. 7a). Actually, CUCPT8m improved erythroid differentiation (as indicated by CD235a expression) and proliferation of *RPS19*$^{+/-}$ cells (Fig. 7b–d). Besides, increased p-STAT5 after CUCPT8m treatment in WT and *RPS19*$^{+/-}$ cells was detected (Fig. 7e, f).

Encouraged by this observation, we lastly investigated the effect of TLR8 inhibition on erythropoiesis of BM-derived CD34$^+$ cells from RP-mutant DBA patients. CUCPT8m remarkably improved the erythroid differentiation of cells from the *RPS19*-mutant patient unresponsive to glucocorticoid (Fig. 7h). Additionally, both CUCPT8m and CUCPT9a (another TLR8 inhibitor) increased cell number of *RPL5*-mutant patients (Fig. 7i). These results suggest that TLR8 is the potential target for improving physiological and pathological erythropoiesis of DBA.

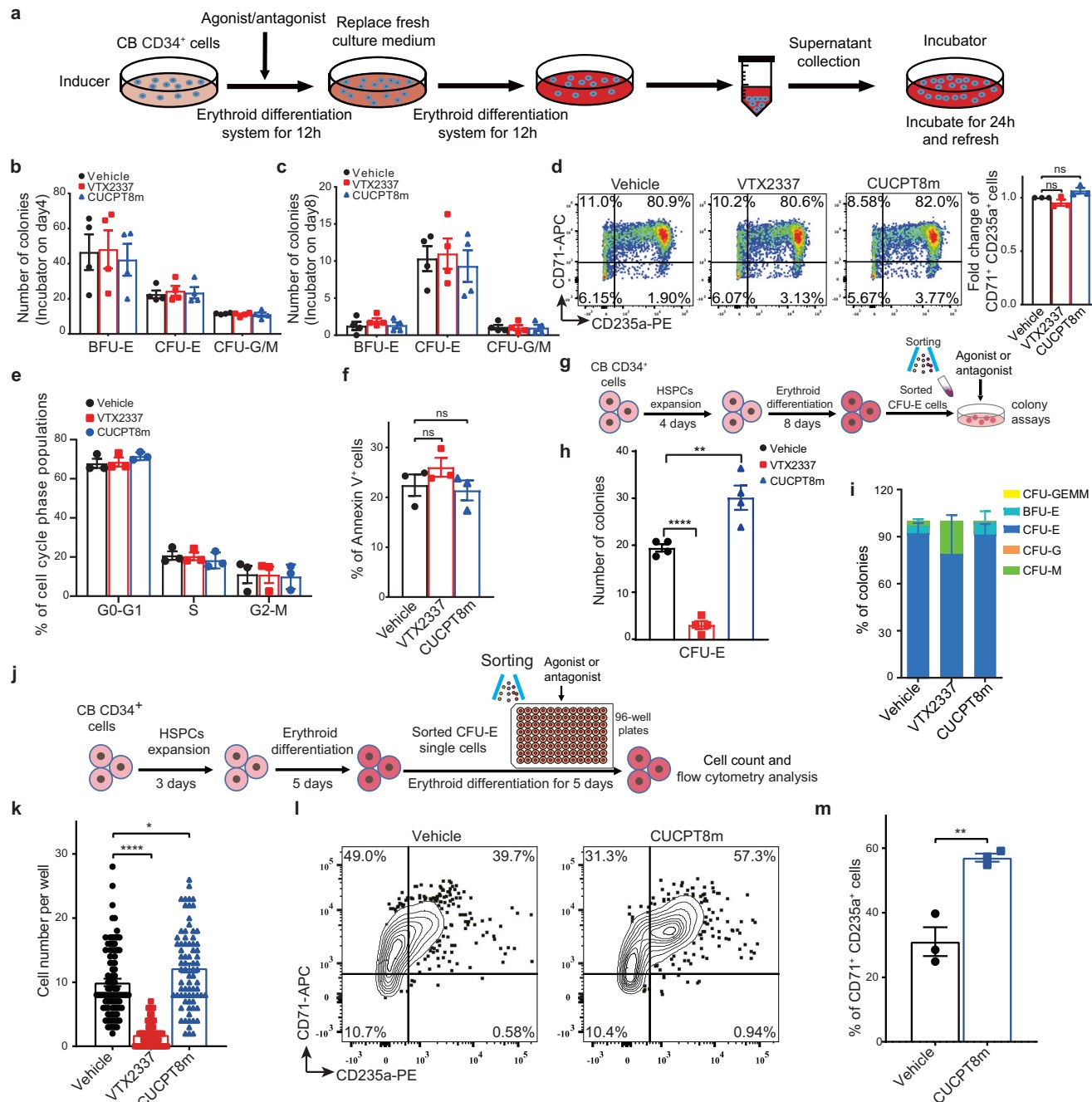

**Fig. 5 | TLR8 regulates erythropoiesis in erythroid cell-intrinsic manner.**
**a** Schematic illustrating the inducer-incubator assay workflow. Numbers of colonies generated from incubator cells collected on days 4 (**b**) and 8 (**c**) of erythroid differentiation (n = 4 biologically independent experiments). **d** Representative flow cytometry (left) showing the percentage of CD71⁺CD235a⁺ erythroid precursor cells of incubator cells on day 11 of erythroid differentiation. Statistics of the fold changes are shown on the right (n = 3 biologically independent experiments). **e** Cell cycle analysis of incubator cells on day 11 of erythroid differentiation (n = 3 biologically independent experiments). **f** Apoptosis of incubator cells (annexin-V⁺ cells) on day 11 of culture (n = 3 biologically independent experiments). **g** Schematic illustrating experimental design. After 8 days of erythroid differentiation, FACS-sorted CFU-E cells were harvested and subjected to colony-forming assays with 10 μM VTX2337 or 10 μM CUCPT8m treatment. **h** Numbers and **i** percentages of colonies generated from FACS-sorted CFU-E cells under the indicated conditions

(n = 4 biologically independent experiments). **j** Schematic illustrating experimental design. Single CFU-E was sorted and seeded into 96-well plates on day 5 of erythroid differentiation. These purified CFU-E cells were further induced differentiation toward erythroid lineage for 5 days with 10 μM VTX32337 or 10 μM CUCPT8m treatment. **k** Cell number on day 5 of erythroid differentiation upon vehicle, 10 μM VTX2337 or 10 μM CUCPT8m treatment (n = 88 for vehicle, n = 65 for VTX2337, n = 75 for CUCPT8m from biologically triplicate). The representative flow cytometry (**l**) showing percentages of CD71⁺CD235a⁺ erythroid cells after treatment with vehicle or 10 μM CUCPT8m for 5 days during erythroid differentiation and its statistics (**m**) (n = 3 biologically independent experiments). Data are mean ± SEM; statistical significance was determined using the unpaired two-tailed Student's t-test. *P < 0.05, **P < 0.01, ****P < 0.0001, ns = not significant. Source data are provided as a Source Data file.

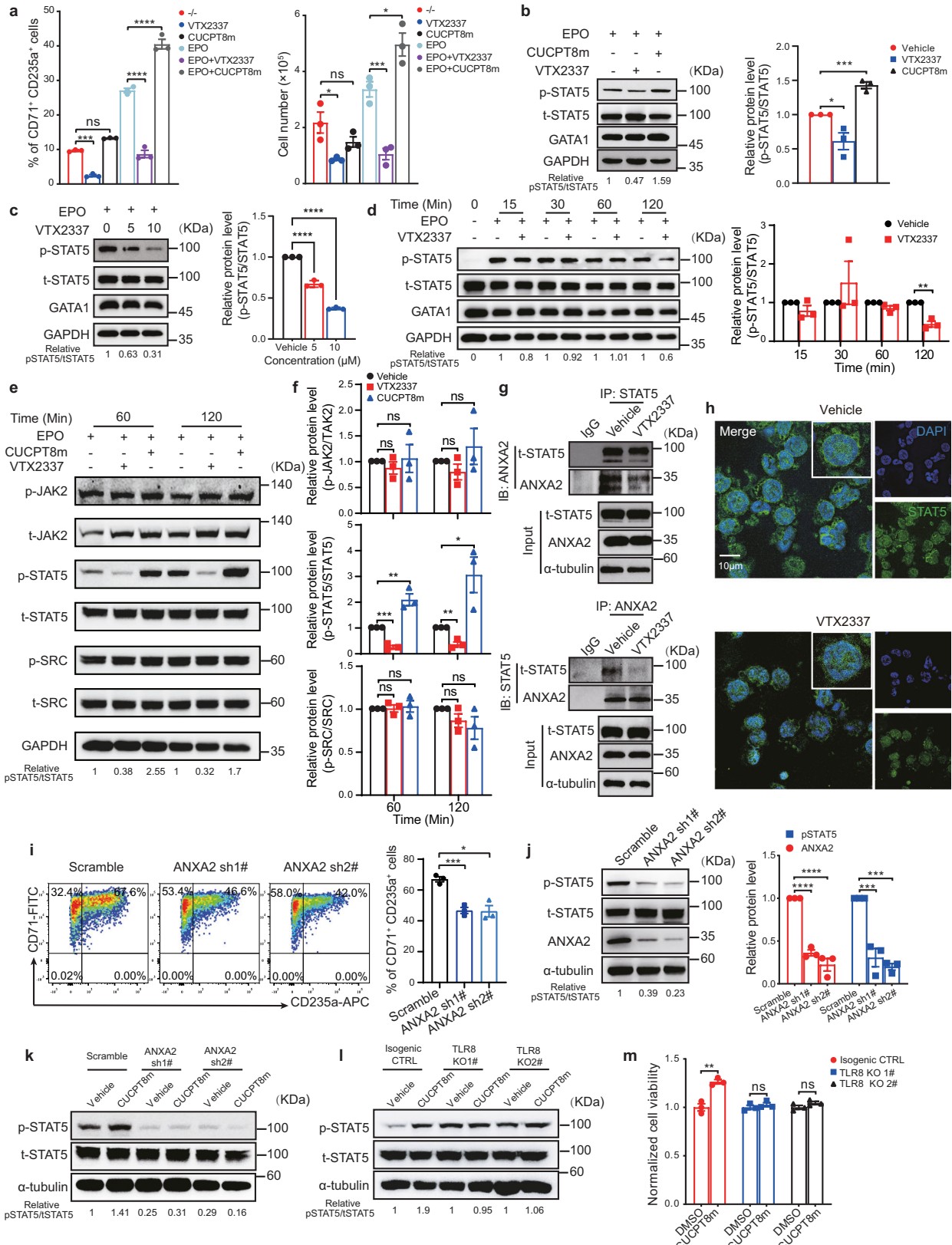

## Discussion

By dissecting the diseased pedigree of the proband with inherited non-hemolytic anemia, we first reported that *TLR8* was the novel causative gene of these diseases. Further studies unraveled TLR8 was expressed in erythroid cells and its activation impaired erythropoiesis from erythroid progenitor stage. Mechanistic investigation showed that TLR8 impaired erythropoiesis by interfering with ANXA2-mediated plasma membrane localization of STAT5 (Fig. 8). Further study showed that TLR8 inhibition improved erythropoiesis of CD34+ HSPCs from healthy donors and RP-mutant DBA patients. Thus, TLR8 potentially served as the novel target for enhancing erythropoiesis under physiological and pathological conditions.

**Fig. 6 | TLR8 impairs erythropoiesis by interfering with the ANXA2-mediated plasma membrane localization and activation of STAT5. a** The percentage of CD71$^+$CD235a$^+$ cells and cell number from days 4 to 8 of erythroid differentiation (CB-derived CD34$^+$ cells). **b** Immunoblotting of STAT5 phosphorylation in cells treated with VTX2337 or CUCPT8m from days 0 to 11 during erythroid differentiation and the quantitation of relative p-STAT5 levels (n = 3 biologically independent experiments). **c** Immunoblotting showing the effects of increasing VTX2337 dosage on STAT5 phosphorylation in HuDEP2 cells and the quantitation of relative p-STAT5 levels (n = 3 biologically independent experiments). **d** Immunoblotting (left) showing the effects of increasing duration of 5 μM VTX2337 on p-STAT5 phosphorylation in HuDEP2 cells and the quantitation of relative p-STAT5 levels (n = 3 biologically independent experiments). **e** Immunoblotting of the indicated proteins of primary erythroid cells on day 6 of erythroid differentiation treated with 10 μM VTX2337/CUCPT8m (n = 3 biologically independent experiments) and **f** the quantitation (n = 3 biologically independent experiments). **g** Co-immunoprecipitation of ANXA2/STAT5 from HuDEP2 cells treated with 10 μM VTX2337 for 2 h (n = 3 biologically independent experiments).

**h** Immunofluorescence of STAT5 in HuDEP2 cells treated with or without VTX2337 for 10 min. Insets are the representative cells. Scale bar = 10 μm. **i** Flow cytometry showing the percentage of CD71$^+$CD235a$^+$ cells in HuDEP2 cells after ANXA2 depletion and the statistics (n = 3 biologically independent experiments). **j** Immunoblotting of p-STAT5 phosphorylation after ANXA2 depletion and the quantitation (n = 3 biologically independent experiments). **k** Immunoblotting of p-STAT5 phosphorylation following 10 μM CUCPT8m treatment in ANXA2-depleted HuDEP2 cells (n = 3 biologically independent experiments). **l** Immunoblotting of p-STAT5 phosphorylation following 5 μM CUCPT8m treatment in TLR8 knockout HuDEP2 cells (n = 3 biologically independent experiments). **m** Normalized cell viability of TLR8 knockout HuDEP2 cells after CUCPT8m treatment for 24 h (n = 3 biologically independent experiments). Data are mean ± SEM; statistical significance was determined using the unpaired two-tailed Student's t-test for two groups comparison or one-way ANOVA for multiple comparisons. *$P < 0.05$, **$P < 0.01$, ***$P < 0.001$, ****$P < 0.0001$, ns = not significant. Source data are provided as a Source Data file.

---

A recent study revealed that TLR8 GOF mutations are associated with bone marrow failure and a broad range of immunodeficiency symptoms[32]. Their study identified two mosaic (p.P432L and p.F494L) and one germline (p.G572F) *TLR8* mutations. By contrast, we identified a novel germline GOF mutation in *TLR8* (p.V434L) in the pedigree that the proband suffered from anemia. Moreover, other patients in this pedigree predominantly manifested hematological disorders. Although the patients in the prior study were susceptible to infection, this was not observed in our study. However, given the distinct clinical phenotypes introduced by distinct *TLR8* GOF mutations, we reasoned that different pattern of TLR8 activation due to different mutations intrinsically coupling with environmental cues promoted the development and progression of TLR8-related diseases.

In our report, the proband exhibited inherited anemia. We showed for the first time that TLR8 was expressed in erythroid cells and its GOF mutation impaired erythropoiesis by disturbing EPO signaling. EPO signaling is crucial for erythroid survival, proliferation, and differentiation[33,34]. In response to EPO, STAT5 forms homodimers and translocates to the nucleus, where it upregulates downstream targets, such as *BCL-xL*[33]. In this study, we found TLR8 activation attenuated while TLR8 inhibition enhanced EPO signaling by manipulating STAT5 phosphorylation. Further, TLR8 acted by disrupting ANXA2-mediated localization of STAT5 at the plasma membrane. By linking TLR8 activity with plasma membrane localization of STAT5, our study provides an additional layer to the regulatory mechanisms of STAT5 and offers novel insights into the regulation of erythroid fate decisions. Nonetheless, how TLR8 signals ANXA2 and thus regulates the interaction between STAT5 and ANXA2 is not elucidated, which is the limitation of this study and deserves further investigations.

Glucocorticoid treatment has been proved to ameliorate the blood transfusion independence in anemic conditions, including certain inherited non-hemolytic anemia[35]. However, there remains patients who are either unresponsive to glucocorticoid or cannot tolerate the side effects. For these patients, novel therapies are needed. Here, we explored the potential clinical implications by assessing the effect of TLR8 inhibition on abnormal erythropoiesis of DBA, one of the major forms of the inherited non-hemolytic anemia. We demonstrated TLR8 inhibition combined with glucocorticoid improved the erythropoiesis in DBA patients. It suggests that targeting TLR8 might help to improve the efficacy or responsiveness of erythroid cells to glucocorticoid in these patients. Nonetheless, to further explore the possible clinical application of TLR8 inhibitors, their dosage should be carefully determined to avoid the possible immune-suppressive side effects, especially when administrated with glucocorticoids.

Additionally, it is interesting to note that the proband, who carrying the GOF TLR8 mutation, became transfusion-independent after glucocorticoid treatment. Prior study has reported that glucocorticoid could restrain the activity of TLR7 and TLR8 in dendritic cells[36]. By linking the glucocorticoid with TLR8, we then speculate that the therapeutic efficacy of the proband might associate with the reduced TLR8 activity and the resultant activation of JAK-STAT signaling pathway in erythroid cells, ultimately promoting erythropoiesis.

As TLR8 is highly expressed in immune cells, it was reasonably to deduce that the microenvironment surrounding erythroid cells would be modified by immune cells carrying *TLR8*$^{V434L}$ and thus contributed to dyserythropoiesis in our case. In concerted with such hypothesis, a very recent study reported that TLR8 impaired the nursing capacity of EBI macrophage and resulted in the compromised erythropoiesis in systemic lupus erythematosus prone mice carrying human TLR8[37]. Nonetheless, the potential extrinsic effect of TLR8 activation on erythropoiesis awaits further investigations.

In sum, we identified a novel GOF mutation in *TLR8* (p.V434L) in a diseased pedigree of the proband with inherited non-hemolytic anemia. Further studies indicated that TLR8 was expressed in erythroid cells and could be a novel regulator of physiological and pathological erythropoiesis which acted through disturbing EPO signaling. Importantly, TLR8 inhibition improved physiological and pathological erythropoiesis. Our findings indicate that TLR8 could serve as the novel therapeutic target for pathological erythropoiesis in diseases.

## Methods

### Collection of human peripheral blood, cord blood, and bone marrow cells

PBMNCs were collected from healthy individuals (including the father, mother, uncle, and grandfather of the proband in this pedigree) and patients (including the proband and his grandmother) and cryopreserved. CB was obtained from the Biobank at the Blood Diseases Hospital; BM samples from the patients (three RP-mutant DBA patients) and healthy donors were also collected at the Blood Diseases Hospital. The written informed consent for the analysis of human samples and the publication of potentially identifiable medical data was obtained from the research participants or their parents/legal guardians. The present study was approved by the Ethical Committee on Medical Research at the Institute of Hematology and Blood Diseases Hospital (Tianjin, China, KT2019090-EC-2, KT2019090-EC-3) and followed the CARE guidelines and the STROBE statement. DBA was diagnosed according to the criteria specified at the Sixth Annual Daniella Maria Arturi International Consensus Conference[38]. Clinical and laboratory features of the proband were assessed (summarized in Fig. 1b, c).

### Chemicals

PMA (CAT# HY-18739), VTX2337 (CAT# HY-13773), and R848 (CAT# HY-13740) were purchased from Medical Chemical Express (MCE).

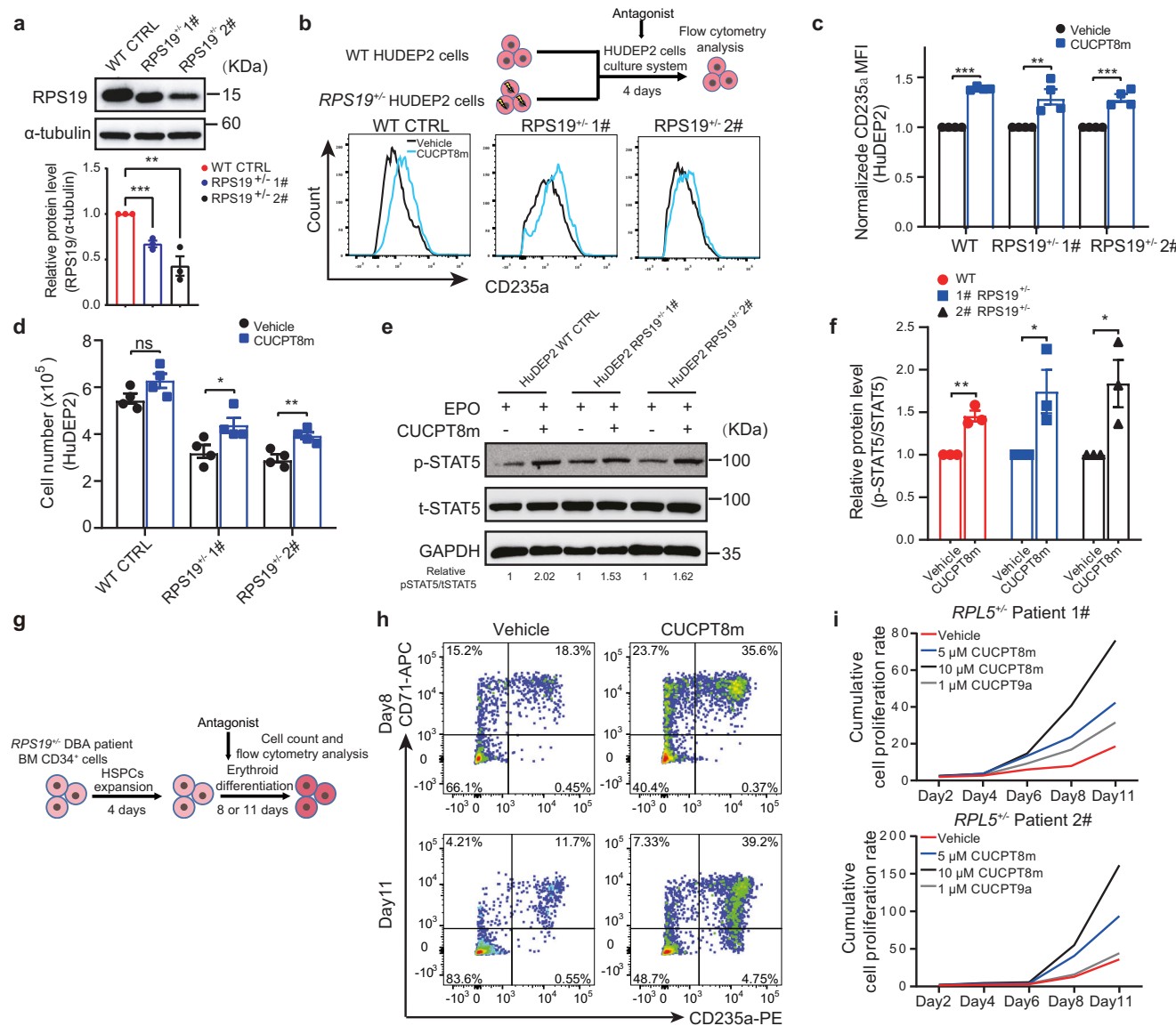

**Fig. 7 | TLR8 inhibition improves the number and differentiation of erythroid cells generated by cell lines and primary cells from healthy donors and DBA patients. a** Representative immunoblotting showing RPS19 levels in WT and *RPS19*[+/-] HuDEP2 cells and the quantitation (n = 3 biologically independent experiments). **b** Schematic illustrating the experimental design for analysis of *RPS19*[+/-] HuDEP2 cell lines. Lower panel shows representative levels of CD235a expression in WT and *RPS19*[+/-] HuDEP2 cells after treatment with 10 μM CUCPT8m for 4 days (n = 4 biologically independent experiments). **c** Quantification of MFI of CD235a expression shown in (**b**) (n = 4 biologically independent experiments). **d** Total cell numbers of WT and *RPS19*[+/-] HuDEP2 cells treated with/without 10 μM CUCPT8m. 2 × 10⁵ cells are seeded accompanied with 10 μM CUCPT8m treatment for 4 days. Total cell number is then counted (n = 4 biologically independent experiments).

**e** Representative immunoblotting showing levels of p-STAT5 and t-STAT5 in WT and *RPS19*[+/-] HuDEP2 cells treated with 3 U/ml EPO alone or in combination with 10 μM CUCPT8m for 2 h and **f** quantification of relative protein expression (p-STAT5/t-STAT5; n = 3 biologically independent experiments). **g** Schematic illustrating erythroid differentiation and analysis of CD34⁺ cells from RP-mutant DBA patient. **h** Flow cytometry results showing erythroid differentiation (days 8 and 11) of BM-derived CD34⁺ cells from a RPS19-mutant DBA patient after consecutive treatment with 10 μM CUCPT8m. **i** Cumulative proliferation curve of BM-derived CD34⁺ cells from two RPL5-mutant DBA patients treated with/without TLR8 inhibitors. Data are mean ± SEM; statistical significance was determined using the unpaired two-tailed Student's t-test. *P < 0.05, **P < 0.01, ***P < 0.001, ns = not significant. Source data are provided as a Source Data file.

---

CUCPT8m (CAT# S4461) and CUCPT9a (CAT# S9730) were purchased from Selleck. Chemicals were dissolved in DMSO and applied according to the manufacturer's instructions.

## Culture of HEK-293T cells

HEK-293T cells were maintained in DMEM (CAT# c11330500BT, Gibco) supplemented with 10% fetal bovine serum (FBS; CAT# 16000-044, Gibco) and 1% penicillin/streptomycin in a humidified incubator at 37 °C with 5% CO₂. Cells were passaged when confluency reached 80–90%.

## Luciferase reporter assays

Briefly, when HEK-293T cells reached 80–90% confluency, they were trypsinized using 0.25% trypsin–EDTA for 1 min at 37 °C. Digestion was stopped by the addition of complete culture medium, and cells were then plated at a density of 2 × 10⁵/well in 24-well plates. When confluency attained 50%, cells were transfected with 1.5 μg GFP or wild-type/mutant TLR8 (WT and mutant TLR8 were synthesized by BGI (Shenzhen, China), expression plasmids were kindly provided by Professor Jiaxi Zhou), 250 ng pNF-κB-TA-luc (CAT# D2207, Beyotime Biotechnology), and 60 ng pRL-TK-luc (CAT# D2760, Beyotime

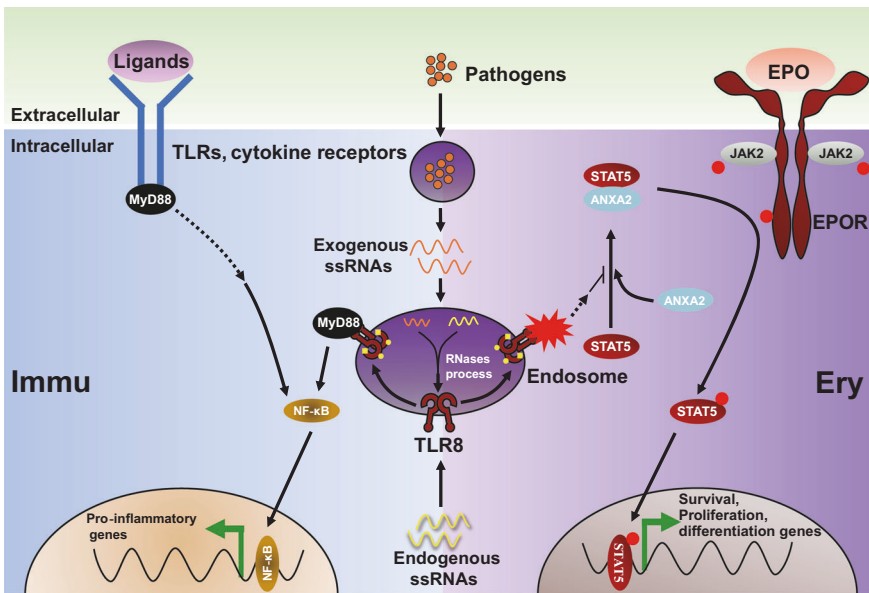

**Fig. 8 | Schematic illustration of our study.** TLR8 in immune cells activates NF-κB signaling upon ligand ligation which is crucial for inflammatory cytokine production. Nonetheless, activation of TLR8 interferes with ANXA2-mediated plasma membrane localization of STAT5 and subsequent activation, ultimately leading to impaired erythropoiesis due to disrupted EPO signaling in erythroid cells.

Biotechnology) plasmids using FuGENE® transfection reagent (CAT# E2311, Promega) in a humidified incubator at 37 °C with 5% $CO_2$. After 24 h, the culture medium was replaced with basic medium supplemented with agonists and incubated for another 12 h. Finally, the cells were lysed and reporter assays were conducted using Dual-Luciferase® Reporter Assay System (CAT# E1910, Promega) according to the manufacturer's instructions. pNF-κB-TA-luc signals were normalized to pRL-TK-luc signals.

**Induction of macrophage lineage differentiation in THP1 cells**
THP1 cells were maintained in RPMI-1640 (CAT# c11875500BT, Gibco) supplemented with 10% FBS and 1% penicillin/streptomycin in a humidified incubator at 37 °C with 5% $CO_2$. Cell density was maintained at $2 \times 10^5$–$2 \times 10^6$/ml. For the induction of differentiation, THP1 cells were treated with 100 ng/ml PMA dissolved in basic RPMI-1640 for 2 days, followed by resting for 3 days in basic RPMI-1640. The differentiated cells were then subjected to treatment and experimental analysis.

**Induction of erythroid lineage differentiation in hESCs**
The male H1 human embryonic stem cells (hESCs) were digested with accutase (CAT# A1110501, Invitrogen) for 1 min at 37 °C to yield single-cell suspension. Then, $1.5 \times 10^5$ cells were seeded in Matrigel-coated 12-well plates and cultured with mTeSR1 (CAT# 85852, STEMCELL Technologies) containing 10 μM Y27632 (CAT# 688000, Calbiochem) to generate hESC colonies. After 48 h, these hESCs colonies were digested with dispase (CAT# 42613-33-2, Solarbio) and cultured in low-attachment 12-well plates using mTeSR1 to form the embryonic bodies (EB). 24 h later, the EBs were induced for hematopoietic differentiation with the basic medium of StemPro-34 (CAT# 10639011, Gibco) supplemented with 1% penicillin/streptomycin, 2 mM L-glutamine (CAT # 25030, Gibco), 1 mM ascorbic acid, 40 mM MTG (CAT# M6145, Sigma) and 150 μg/ml transferrin[39] (CAT# 10652202001, Sigma). During this differentiation process, the basic medium was supplemented with 10 ng/ml BMP4 (CAT# 120-05ET-1000, Peprotech) on day 1; 10 ng/ml BMP4, 5 ng/ml bFGF (CAT# 100-18B-100, Peprotech) on day 2; 10 ng/ml BMP4, 5 ng/ml bFGF, 3 μM CHIR99021 (CAT# SML1046, Sigma) from days 3 to 4; 10 ng/ml BMP4, 5 ng/ml bFGF, 25 ng/ml IGF-1 (CAT#

100-11, Peprotech), 10 ng/ml IL-6 (CAT# 78050.1, STEMCELL Technologies), 5 ng/ml IL-11 (CAT# 200-11, Peprotech) from days 5 to 6; 15 ng/ml VEGF, 5 ng/ml bFGF, 25 ng/ml IGF-1, 10 ng/ml IL-6, 5 ng/ml IL-11, 50 ng/ml SCF (CAT# CYT-255, Prospec), 2 U/ml EPO (CAT# 10064, Prospec) from days 7 to 8; and 15 ng/ml VEGF (CAT# 100-20-100, Peprotech), 5 ng/ml bFGF, 25 ng/ml IGF-1, 10 ng/ml IL-6, 5 ng/ml IL-11, 50 ng/ml SCF, 2 U/ml EPO, 30 ng/ml TPO (CAT# 30018 PeproTech), 10 ng/ml Flt-3L (CAT# 300-19, PeproTech), and 30 ng/ml IL-3 (CAT# I1646, Sigma) from days 9 to 14. Thereafter, CD45⁺ HSPCs were FACS-sorted for either colony-forming assays or induction of erythroid differentiation. Erythroid differentiation was induced in a two-phase culture system. First, cells were proliferated and differentiated in basic medium (StemSpan SFEM medium (CAT# 09650, STEMCELL Technologies) containing 1% BSA (Bovine serum albumin, CAT# A4161, Sigma), 0.0085% β-mercaptoethanol (CAT# M7522, Sigma), 7.7 μg/ml transferrin (CAT# T2252, Sigma)) supplemented with 1 μM dexamethasone (CAT# 9891D2915 Sigma), 100 ng/ml SCF, 3 U/ml EPO, 5 ng/ml IL-3 and 1% penicillin/streptomycin from days 0 to 8. Second, dexamethasone and IL-3 were removed from the basic medium and the cells were further differentiated from days 8 to 14 in a humidified incubator at 37 °C with 5% $CO_2$. Cells were analyzed at the indicated time points.

**Construction of mutant THP1, ES, and HuDEP2 cell lines**
Mutant THP1 cells were constructed by Haixing Biosciences (Suzhou, China). *TLR8$^{V434L}$* THP1 cells were generated using a CRISPR/Cas9-based knock-in strategy. Briefly, sgRNA-1 was designed for the targeted site within *TLR8*. To circumvent the repeated cut at the target site by Cas9, synonymous mutations (p.451L (TTG > CTA), p.452L (CTA > CTC)) were introduced to the mutant TLR8. Cells were electroporated with the sgRNA, Cas9, and oligo donor[40]. Clones were selected and geno-typed by Sanger sequencing.

*TLR8$^{V434L}$* hESCs were constructed by Biocytogen (Beijing, China) using a CRISPR/Cas9-based knock-in strategy. Briefly, sgRNA-2 and sgRNA-3 were designed for the targeted site within *TLR8*. hESCs were electroporated with pCS/Cas9-3G-sgRNA and CL-CYH-018 donor expression plasmids. Clones were selected with puromycin and gen-otyped by Sanger sequencing.

$TLR8^{V434L}$ HuDEP2 cells were generated using a CRISPR/Cas9-based knock-in strategy[41]. Briefly, a sgRNA-4 was designed for the targeted site within *TLR8*. HuDEP2 cells were electroporated with the sgRNA, Cas9, and oligo donor. Clones were selected and genotyped by Sanger sequencing. All the sequences of sgRNAs were listed in Supplementary Table 2.

## ELISA
Supernatants from differentiated THP1 cells with/without agonist treatment were collected, and IL-6 secretion was determined using the ELISA kit (CAT# KIT10395A, Sino Biological) according to the manufacturer's instructions.

## Culture of HuDEP2 cells
HuDEP2 cells were kindly provided by Professor Yukio Nakamura and cultured in StemSpan SFEM supplemented with 50 ng/ml SCF, 1 μg/ml doxorubicin, 1 μM dexamethasone, 3 U/ml EPO, and 1% penicillin/streptomycin in a humidified incubator at 37 °C with 5% $CO_2$[22]. Cell density was maintained between $1 \times 10^5$-$1 \times 10^6$/ml. When cells were subjected to treatment, they were starved with IMDM basic medium for 2 h. Thereafter, cells were treated with cytokine and chemicals dissolved in IMDM basic medium for the indicated time. Then cells were analyzed.

## Cell viability assay
$1 \times 10^4$ HuDEP2 cells/100 μl culture medium were seeded to wells of 96-well plates with the treatments for 24–48 h. The reaction solution of MTS (CAT# G3582, Promega) was mixed with the supernatant of the treated cell in a humidified incubator at 37 °C with 5% $CO_2$ for 2 h. OD values at 490 and 670 nm (background) were determined respectively. The cell viability was obtained by calculating the difference between two values.

## Structural modeling
The published structure (PDB ID: 3W3K) was chosen for agonist-binding state representation. Two protomers in a TLR8 dimer were shown as cartoons with the two chains colored in forest green and goldenrod, respectively. V434 was shown in ball stick and the agonist was shown as orchid stick. p.V434L mutation was modeled by using COOT[42]. Structural images were generated using Chimera X[43].

## Induction of erythroid differentiation in CB/BM-derived CD34$^+$ HSPCs
Isolation, expansion, and erythroid differentiation of human CB/BM-derived CD34$^+$ HSPCs were conducted[23,24]. Briefly, CD34$^+$ cells were isolated using anti-CD34 magnetic beads (CAT#130046703, Miltenyi Biotec). Erythroid differentiation was induced in a three-phase culture system. First, CB-derived CD34$^+$ cells were expanded in StemSpan SFEM medium (CAT# 09650, STEMCELL Technologies) supplemented with 100 ng/ml SCF, 100 ng/ml IL-6, 100 ng/ml TPO, 100 ng/ml Flt-3L, and 1% penicillin/streptomycin for 3 days in a humidified incubator at 37 °C with 5% $CO_2$. Second, cells were allowed to proliferate and differentiate in basic medium (IMDM (CAT# 13390, Sigma) containing 4 mM L-glutamine, 40 μg/ml inositol (CAT# I5125, Sigma), 10 μg/ml folic acid (CAT# F7876, Sigma), 160 μM 1-thioglycerol (CAT# F7876, Sigma), 90 ng/ml ferrous nitrate (CAT# 8508, Sigma), 900 ng/ml ferrous sulfate (CAT# F8633, Sigma), 20% BIT (CAT# 9500, STEMCELL Technologies), and 1% penicillin/streptomycin). From days 0 to 8, the medium was further supplemented with 1 μM hydrocortisone (CAT# H2270, Sigma), 100 ng/ml SCF, 3 U/ml EPO, 5 ng/ml IL-3, and 1% penicillin/streptomycin. Third, hydrocortisone and IL-3 were omitted from the basic medium and the cells were further differentiated during days 8–14 in a humidified incubator at 37 °C with 5% $CO_2$. The protocol for inducing erythroid differentiation in adult BM-derived CD34$^+$ cells was essentially the same, except that hydrocortisone was replaced by dexamethasone. Cells were cultured at a density of $1 \times 10^5$–$1 \times 10^6$/ml and analyzed on days 4, 8, 11, and 14 of culture.

## Flow cytometry and cell sorting
Cells were washed with PBS and centrifuged at $300 \times g$ for 5 min. The supernatant was then discarded and the cells were resuspended in 100 μl FACS buffer (PBS with 2% FBS + 2 mM EDTA). The antibodies applied in this study were as follows: eFluo 450-conjugated CD3 (CAT# 48003742, eBioscience, OKT3), eFluo 450-conjugated CD4 (CAT# 48004942, eBioscience, RPA-T4), eFluo 450-conjugated CD14 (CAT# 48014942, eBioscience, 61D3), eFluo 450-conjugated CD19 (CAT# 48019942, eBioscience, HIB19), PE-Cy7-conjugated CD34 (CAT# 25034942, eBioscience, 4H11), BV605-conjugated CD123 (CAT# 306026, Biolegend, 6H6), FITC-conjugated CD71 (CAT# 11071942, eBioscience, OKT9 (OKT-9)), APC-conjugated CD235a (CAT# 551336, BD Bioscience, GA-R2), PerCP/Cy5.5-conjugated CD36 (CAT# 561536, BD Bioscience, CB38), PE-conjugated CD45 (CAT# 555483, eBioscience, HI30), APC-conjugated CD41a (CAT# 559777, BD Bioscience, HIP8), PE-conjugated TLR8 (MA516194, Thermo Fisher, 44C143), PE-isotype IgG for TLR8 (CAT# 555749, BD Bioscience, MOPC-21), PE-conjugated CD235a (CAT# 12998782, eBioscience, HIR2 (GA-R2)), APC-conjugated CD71 (CAT# 17071942, eBioscience, OKT9 (OKT-9)), and APC-conjugated CD11b (CAT# 101212, Biolegend, M1/70). Cells were incubated with antibodies (diluted at 1:100) at 4 °C in the dark for 30 min and washed by and resuspended in FACS buffer for flow cytometric analysis ($1-5 \times 10^5$) or sorting ($1-5 \times 10^7$). Erythroid precursors were defined as CD71$^+$CD235a$^+$ cells.

## Intracellular protein staining for flow cytometry
Cells were washed by and resuspended in FACS buffer. Then, they were incubated with antibodies against surface markers of BFU-E/CFU-E, either before or after fixation, at 4 °C in the dark for 30 min (CAT#557870, BD Biosciences). After being washed in PBS, cells were permeabilized in permeabilization buffer (CAT#558050, BD Biosciences) according to the manufacturer's instructions and stained with anti-TLR8 or isotype IgG antibody at 4 °C in the dark for 30 min. After further washes in PBS, the cells were incubated with the remaining antibodies against BFU-E/CFU-E surface markers at 4 °C in the dark for 30 min. Following further washes in PBS, cells were resuspended in PBS and analyzed by flow cytometry.

## Inducer-incubator assay
CM was harvested from inducer CD34$^+$ cells stimulated with VTX2337/CUCPT8m. The response of incubator CD34$^+$ cells to potentially altered cytokine levels in the CM were then determined. Briefly, inducer cells were pre-incubated with erythroid differentiation medium with VTX2337/CUCPT8m for 12 h. After the cells were centrifuged at $300 \times g$ for 5 min, the supernatant was replaced with fresh erythroid differentiation medium without VTX2337/CUCPT8m, and CM production proceeded for 12 h. The CM was then applied to the incubator cells and replaced with fresh CM after 24 h. The incubator cells at specific time point were subjected to analysis and the multiplex cytokine/chemokine assay of CM on days 4 and 8 was determined.

## Wright-Giemsa staining
The cells to be stained ($5 \times 10^4$) were washed by and resuspended in PBS. The cells were then subjected to cytospin at $800 \times g$ for 5 min, followed by fixation with 100% methanol at room temperature (RT) for 1 min. Staining was performed using the Wright-Giemsa Stain kit (CAT# BA4017, BaSO Biotechnology) according to the manufacturer's instructions. Images were captured at 100× magnification.

## Immunofluorescence
After being washed and resuspended in PBS, cells were subjected to cytospin at $800 \times g$ for 5 min and fixed in 4% paraformaldehyde at RT

for 15 min. Cell membrane permeabilization was carried out using 0.1% (v/v) Triton X-100/0.5% BSA at RT for 15 min. The cells were then washed in PBS, and 1% BSA was applied at RT for 1 h to block non-specific antigens. Next, cells were incubated with primary antibodies (anti-EEA1, CAT# ab70521, Abcam, 1G11; anti-TLR8, CAT# bs-8684R, Bioss; isotype IgG for TLR8, CAT# PP64, Sigma Aldrich; anti-STAT5a, CAT# SC-271542, Santa Cruz, C-6) at 4 °C overnight. Following washes in PBS, the cells were incubated with secondary antibodies (diluted 1 in 200) at RT for 2 h (488 donkey anti-mouse IgG (H + L), CAT# A21202, Thermo Fisher; 594 donkey anti-rabbit IgG (H + L), CAT# A21207, Thermo Fisher). After washes in PBS, the slides were mounted in antifade mounting medium with DAPI (CAT# P0131, Beyotime Biotechnology). Images were captured using an inverted confocal microscope.

## Immunoblotting
Briefly, cells were washed with PBS and centrifuged at $300 \times g$ for 5 min. The cell pellets were lysed using RIPA lysis buffer (CAT#R0010, Solarbio) supplemented with protease and phosphatase inhibitors on ice for 15 min and then centrifuged at $13,000 \times g$ for 15 min. Protein concentration was determined using bicinchoninic acid reagent (BCA, CAT# 23227, Thermo Fisher) and the cell lysates were subjected to SDS–PAGE. Antibodies for immunoblotting were as follows: anti-JAK2, (CAT# 3230, CST, D2E12); anti-p-JAK2 (CAT# 3771, CST, Tyr1007/1008); anti-STAT5 (CAT# 94205, CST, D2O6Y); anti-p-STAT5 (CAT# 9359, CST, C11C5); anti-SRC (CAT# 110971-AP, Proteintech); anti-p-SRC (CAT# 6943, CST, D49G4); anti-α-tubulin (CAT# ab11304, Abcam, B-5-1-2); anti-β-actin (CAT# 3700, CST, 8H10D10); anti-GAPDH (CAT# 600041-Ig, Proteintech, 1E6D9); anti-RPS19 (CAT# SC-100836, Santa Cruz, WW-4); anti-TLR8 (CAT# 11886, CST, D3Z6J); anti-GATA1 (CAT# ab181544, Abcam, EPR17362); anti-ANXA2 (CAT# 11256-1-AP, Proteintech); anti-PGAM5 (CAT# 28445-1-AP, Proteintech); anti-sodium potassium ATPase (CAT# ab76020, Abcam, EP1845Y), HRP-conjugated goat anti-mouse IgG (H + L) (CAT# SA00001-1, Proteintech); HRP-conjugated goat anti-rabbit IgG (H + L) (CAT# SA00001-2, Proteintech). Blots were visualized using chemiluminescence (CAT#36222ES76, Yeasen), and images were analyzed using ImageJ software. All uncropped blots were supplied in the Data Source file and Supplementary Information.

## Colony-forming assays
Colony-forming assays were performed using MethoCult™ semi-solid medium (CAT# 04435, STEMCELL Technologies). Briefly, basic medium containing cells was mixed with the semi-solid medium and vortexed vigorously for 1 min. 500 µl mixed medium was then gently aliquoted into 12-well plates and incubated in a humidified incubator at 37 °C with 5% CO₂. CFU-E colonies were counted on day 7, while BFU-E colonies and other CFUs were counted on day 14 after seeding. Colonies were identified according to the manufacturer's instructions.

## Apoptosis analysis
The cells for analysis ($1 \times 10^5$) were washed with FACS buffer and centrifuged at $300 \times g$ for 5 min. They were then resuspended and prepared using an apoptosis detection kit (CAT#556547, BD Bioscience) according to the manufacturer's instructions. For apoptosis assay of erythroid subpopulation in hESCs, CD71⁺CD235a⁺cells were labeled with antibodies against CD71, CD235a, and annexin V-FITC and analyzed. For apoptosis assay of CFU-E, cells were stained with antibodies against CD123, CD235a, CD34, and CD36 followed by annexin V-FITC staining and CD123⁻, CD235a⁻, CD34⁻, CD36⁺ cells were analyzed.

## Co-immunoprecipitation
A total of $1 \times 10^7$ cells were washed with PBS and lysed in native lysis buffer (CAT# R0030, Solarbio) supplemented with protease and phosphatase inhibitors on ice for 15 min. Protein concentration was

determined using BCA, then 1 mg protein in 1 ml lysis buffer was incubated with antibodies including anti-STAT5 (CAT# 94205, CST, D2O6Y) and anti-ANXA2 (CAT# 11256-1-AP, Proteintech), with gentle rotation at 4 °C overnight. Protein G magnetic beads (CAT# 100040, Thermo Fisher) were washed in lysis buffer before being added to the antigen–antibody complex and incubated at 4 °C for 4–6 h. Following washes in lysis buffer, the incubated beads were boiled in loading buffer containing β-mercaptoethanol for 10 min and subjected to SDS–PAGE. The blots were visualized by chemiluminescence.

## In vitro phosphatase assay
Recombinant WT and H105A mutant PGAM5 proteins were kindly provided by Professor Guo Chen (Nankai University, Tianjin). Phosphorylated STAT5 was immunoprecipitated using an anti-p-STAT5 antibody (CAT# 9359, CST) as described above. Briefly, immunoprecipitated p-STAT5 (extracted from $1 \times 10^7$ cells) was incubated with or without 0.5 µg WT/mutant PGAM5 in a phosphatase reaction system (50 mM imidazole [pH = 7.2], 0.2 mM EGTA, 0.02% β-mercaptoethanol, and 0.1 mg/ml BSA) at 30 °C for 30 min. Then, the samples were denatured and analyzed by immunoblotting.

## Quantitative RT-PCR (qRT-PCR)
Cells ($2 \times 10^5$) were lysed in TRIzol reagent (CAT# 15596018, Thermo Fisher) and vortexed vigorously. Following the addition of chloroform, the lysates were incubated at RT for 5 min. After centrifugation at $12,000 \times g$ for 10 min, the upper layer containing the RNA was aspirated and precipitated using isovolumic isopropanol. The RNA was washed with 75% ethanol and dissolved in H₂O. RNA concentration was determined by NanoDrop 2000 spectrophotometer, and 0.5–1 µg RNA was subjected to reverse transcription using Hifair® II 1st Strand cDNA Synthesis SuperMix according to the manufacturer's instructions (CAT# 11141ES60, Yeasen). The cDNA was then analyzed by qPCR using Hieff UNICON® Universal Blue qPCR SYBR Green Master Mix according to the manufacturer's instructions (CAT# 11184ES25, Yeasen). The sequence of each primer pair was listed in Supplementary Table 2.

## Plasmid construction, lentivirus production, and infection
Sequences of shRNA targeting ANXA2 were obtained from Sigma Aldrich and cloned into the pSIH-H1 shRNA lentiviral expression cassette. The shRNA sequences were listed in Supplementary Table 2. After being infected for 48 h, GFP⁺ HuDEP2 cells were sorted and cultured for another 48 h and subjected to treatment and analysis.

## Cell cycle analysis
Cells were fixed by cold ethanol followed by incubation at 4 °C for 2 h. After being washed in PBS, cells were permeabilized and stained using 200 µl PI staining buffer (1% BSA + 100 µg/ml RNase A (CAT# RT40502, Tiangen Biotechnology) + 50 µg/ml propidium iodide (CAT# C0080, Solarbio) + 0.1% Triton X-100) in the dark at 37 °C for 30 min. Cells were then subjected to flow cytometry.

## Isolation of plasma membrane protein
$1 \times 10^7$ HuDEP2 cells were incubated with 3 U/ml EPO with/without 10 µM VTX2337 at RT for 10 min. Afterward, cells were collected and plasma membrane protein was isolated with the Plasma Membrane Protein Isolation and Cell Fractionation kit (CAT# SM-005, Invent Biotechnologies) according to the manufacturer's instructions followed by immunoblotting analysis.

## Statistics and reproducibility
All representative images were repeated independently at least for three times except for Fig. 1c (repeated for two times with the sample from proband). Statistical significance was determined using the unpaired two-tailed Student's t-test for two-group comparison or one-way ANOVA for multiple comparisons. All statistical analyses were

# Article

performed using GraphPad Prism 8. *P* < 0.05, **P* < 0.01, ***P* < 0.001, ****P* < 0.0001, ns = not significant. The exact *P* values produced in the figures were provided in Source Data file.

### Reporting summary

Further information on research design is available in the Nature Portfolio Reporting Summary linked to this article.

## Data availability

All sequencing data generated in this study have been deposited in the Genome Sequence Archive in National Genomics Data Center, China National Center for Bioinformation/Beijing Institute of Genomics, Chinese Academy of Sciences that are publicly accessible (https://ngdc.cncb.ac.cn/gsa-human/s/tmVGhn9D). The processed and generated data in this study are provided in the Supplementary Information and Source Data files which are provided with this paper. Source data are provided with this paper.

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

## Acknowledgements

We thank Tao Jiang for technical assistance of TLR8 structure modeling; Shangda Yang for technical assistance of CoIP samples preparation for mass spectrometry analysis. This study was supported by grants from the National Key Research and Development Program of China (2022YFA1103503 to L.S.), the National Natural Science Foundation of China (82225003 to L.S., 82270144 to X.Z., 82100152 to J.T., 82300141 to J.G., 22137004 to H.Y., 21825702 to H.Y.), China Postdoctoral Science Foundation funded project (2023M740321 to J.L.), the CAMS Innovation Fund for Medical Sciences (2021-I2M-1-040 to L.S., 2021-I2M-1-041 to X.Z., 2021-I2M-1-073 to J.T.), Haihe Laboratory of Cell Ecosystem Innovation Fund (22HHXBSS00017 to L.S., HH24KYZX0005 to L.S.), State Key Laboratory of Experimental Hematology Research Grant (Z23-10 to L.S.), Tianjin Municipal Science and Technology Commission Grant (23ZXRKSY00010 to L.S.), Beijing Outstanding Young Scientist Program (BJJWZYJH01201910003013 to H.Y.) and Beijing Advanced Innovation Center for Structural Biology (20151551402 to H.Y.), Special Research Fund for Central Universities, Peking Union Medical College (3332023059 to X.L.). We also thank members of the experimental platform of IHCAMS and the Division of Pediatric Blood Diseases Center.

## Author contributions

L.S. and J.L. designed and analyzed the experiment. L.S. and X.Z supervised the project. J.L., J.G., L.Z., J.W., Y.L., B.W., Ding W., Y.M., X.L., D.W., and G.G. performed the experiment. P.Z. and P.W. performed the bioinformatic analysis. X.Z., Y.W., and B.S. collected the samples and information of patients. N.A. and G.C. aided in dephosphorylation ex vivo of p-STAT5. X.M. performed the structural remodeling. M.G. and H.Y. provided the CUCPT8m. R.K. and Y.N. provided HuDEP2 cell lines and the culture protocol. J.T. and J.H.L. provided suggestive advice.

## Competing interests

The authors declare no competing interests.
