## [Peer Review File · Nature Communications]

REVIEWER COMMENTS

Reviewer #1 (Remarks to the Author):

The manuscript of Liang J and colleagues reports the discovery of a homozygous gain-of-function mutation in TLR8 gene in a patient with anemia 3 weeks after birth resembling a DBA phenotype. This study based on a case-report and first description of a mutation related to hereditary anemia aims at understanding the role of TLR8 in the control of erythropoiesis. Although number of experiments have been done using cell line models and pharmacological modulation of normal erythropoiesis with agonist and antagonist of TLR8, there are several concerns with the conclusions of experiments

Major comments

1. Regarding the case-report, some additional parameters are needed to better characterize the anemia: MCV and inflammatory parameters at diagnosis, Hb level over time and under treatment with steroids (please provide curves), red blood cell transfusion needs before and after introduction of steroids. Compared to DBA linked to heterozygous mutations in ribosomal protein genes, did the proband exhibit bone malformations? Do the authors consider the proband affected by a severe disease? Because the infant presented a homozygous mutation and the mother a heterozygous mutation, the father is expected having a heterozygous mutation: what is the Hb level of the mother and the father? Have they any symptoms?
2. Regarding the West syndrome, such neurological disorders could be improved by steroids often used as first line therapy. Did the infant less affected by repeated spasms after starting the steroids? Did the authors identify one of the mutations reported in the West syndrome (ARX, STXBP1, STK9...) by whole genome sequencing?
3. TLR8 has been reported to be overexpressed in FANCC^{-/-} cells and induced inappropriate production of TNF- α (Vanderwef et al, Blood 2009). It is not clear in this manuscript whether the mutation V434L is associated or not with an increased expression level of TLR8 in THP1-mutated cells?
4. The authors used CRISPR-cas9 strategy to edit the mutation in at least to cell models the monocytic cell line THP1 and the human ESC. They apparently used the parental cells as controls. To better establish the specificity of V434L mutation, experiments must be performed with the isogenic THP1 and hESC V434V cells must be shown.
5. Figure 2: The function of V434L mutant is assessed in monocytic THP1 cell line. The choice of this cell line is curious as the patient presented with anemia. Why do the authors used this cell line? Furthermore, they show that THP1 V434L cells in differentiation conditions upon PMA secrete IL6 at significantly higher levels than the WT THP1. This effect could be related either to the mutation suggesting an extrinsic mechanism of control of hematopoiesis, or to an off-target effect of the CRISPR-Cas9 strategy.
6. What is the effect of the mutation in hESC on the granulomonocytic differentiation? Based on the results obtained in THP1 cells, the granulomonocytic differentiation of hESC is expected to be induced by

the mutation and modulated by the agonist/antagonist of TLR8? Again the hESC V434V cells must be used as controls.

7. Fig. 3g: the effect of VTX2337 treatment on hESC V434V and hESC must be shown.

8. As well as the expression of TLR8 was followed along normal erythroid differentiation by western blot (Fig. 4c), it could be interesting to follow the expression of TLR8 along erythroid differentiation upon treatment with VTX2337 or CUCPT8 by flow cytometry.

9. Fig. 4e-f: the authors purified BFU-E and CFU-E nicely to assess the expression of TLR8. The antibody used to visualize TLR8 recognizes an intracellular domain of the protein making cell permeabilization mandatory for flow cytometry. Additional controls in cell lines are needed to ascertain the specificity of TLR8 antibody labelling. Furthermore, the western blots to TLR8 must be shown entirely at least once in cell line (as a suppl figure).

10. Ext Fig3a and Fig. 4: the authors stated that VTX2337 hampered the differentiation of CD34+-derived erythroid progenitors and precursors and induced apoptosis while CUCPT8 did the contrary. However, they referred to Ann V+ cells which include dead AnnV+ 7-AAD+ cells and early apoptotic AnnV+ 7-AAD- cells. In addition, Ext Fig 3a clearly indicates that VTX 2337 at 20 microM induced cell death (not apoptosis). Altogether this suggests that the decrease of CD235a positive cells induced by VTX2337 at 10 microM is rather due to the increase of cell death from 5 – 8% rather than a blockade of differentiation. To better address these points, the authors must show apoptotic cell percentage (AnnV+ 7-AAD-).

11. Fig. 5: In this interesting experiment, the authors collected conditioned media in control, VTX2337 and CUCPT8 conditions. As endpoint, the authors quantified colonies far from the contact with CM. It should be interesting to perform multiplex cytokine/chemokine assay to address the impact of TLR8 targeting more directly.

12. Fig. 4i-w and Fig. 6: CUCPT8 did not affect cell viability, induced differentiation through an EPO-dependent mechanism implicating STAT5 localization at plasma membrane (shown by IF) due to its interaction with annexin A2. It could be interesting to perform cell fractionation to isolate plasma membrane and confirm the relocation of pSTAT5 to the membrane with an additional method.

Minor comments

Fig 6b: why is there two bands of GATA1 with vehicle and VTX2337 and not with CUCPT8?

Fig 6c-d: the time and dosage dependent decrease of pSTAT5 upon VTX2337 is not obvious. How was the cell viability after 2 hours of VTX2337?

Ext data Fig 4c: increased p-STAT5 avec 2 hours of EPO stimulation is not expected. Rather EPO signaling turns off after 30 min upon negative regulatory signals mediated by phosphatases and ubiquitylation. Is it related to the HUDEP2 cells?

Fig 6g: WB annex A2 of input is lacking.

Fig 6k: the control with shSCR is lacking

Reviewer #2 (Remarks to the Author):

The paper by Liang et al describes the discovery of a novel germline gain-of-function mutation in the gene encoding LTR8 in a child with anemia. These discovery lead Liang et al unveil LTR8 as a new regulator of erythropoiesis.

The phenotype of the proband is strong and the genetic approach that lead to the discovery of the novel mutation is robust. The hypothesis that the mutated gene is responsible for the anemic phenotype of the proband is supported by a well-organized experimental approach which involves the analyses of the effects of the mutated gene in several surrogate models of human erythroid differentiation.

I have a general clinical comment and several specific comments on the data presented in the Figures and on their interpretation.

General comment

The similarity of the anemia expressed by the proband with the macrocytic anemia which characterizes Diamond-Blackfan Anemia (DBA) is thin. The authors should provide strongest diagnostic data by moving the data provided in Table S1 in the main manuscript and by presenting photographs of the bone marrow biopsy. Alternatively, they should remove DBA from the title and mention the possibility that the proband had a novel form of DBA in the discussion. In this case, the introduction should also be revised to briefly overview all the inherited forms of anemia, including DBA. It should be noted that the paper by Kim et al (ref 4) does not mentioned that the inherited anemia induced by the EPO-R mutation is DBA, as stated by the authors.

The fact that the anemia of the proband does not recapitulate DBA does not reduce the merit of the discovery of the regulatory role exerted by LTR8 in erythropoiesis reported by this paper. In this regard, it is tempting to speculate that activation of LTR8 observed during opportunistic infections and necessary to boost the immune-response of the patients may contribute to their anemia. The paper surely include data to sustain this claim.

Specific Comments.

Figure 1. The proband is hemizygous not homozygous as stated. The gene is on chromosome X and males have only one X and may not be homozygous. What does it mean: IV. No limit to mother? Please clarify. Please correct mother (not mather) in the Figure. The capture should clarify how the proband is indicated in the pedigree.

Figure 2. The rationale to investigate the activity of TLR8 on the activation of NF-kB should be provided. Panel a. The experimental design above the bar graph should clarify what the luciferase reporter is reporting (NF-kB activity?). Panel g: are the levels of IL-6 released by the differentiated TLF8V434L cells upon stimulation significantly greater than at baseline? The differentiated cells may be releasing more IL-6 in response to VTX2337 but it is debatable that they respond to R848. Panel h: the relevance of the data presented in this panel is unclear. The structural difference between the wild-type and mutated protein is not evident. The two proteins appear structurally similar. Two tailed Student's t-test is not the appropriate statistical method to analyze data sets which contain more than two groups. The data should be analyzed with a statistical method for multiple comparisons. This comment applies also to the statistical analyses of the data included in other Figures. The presentation of Panel b should be

prioritized before that of panel a.

Take home message of Figure 2: Since un-differentiated and differentiated wild type THP1 cells do not produce IL-6 in response to the agonists, more than a gain of function, the TLV8V434L mutation appears to confer to the protein a novel function. This would be consistent with the effects of the mutations in EKLK which have been described to cause anemia by “acquisition of a new function” rather than by “gain of function”.

Figure 3: The data presented in this Figure are not robust. Panel c. The Y axes should indicate the number of cells plated. The number of cells plated was too low to allow robust assessments of the number of colonies. Differences between 4 or 1-2 BFU-E per plate are not meaningful. The experiment should be repeated plating greater number of cells. As presented now these data do not support the conclusion that TLR8V434L halts the erythroid differentiation from the BFU-E to the CFU-E stage. It is interesting that in one case, there is a significant increase in the number of CFU-GM-derived colonies. This increase is consistent with the data on the macrophage activation presented in Figure 2 and could be real. This result is also consistent with the greater WBC counts of the proband. Do DBA patients have a monocyte phenotype? To be discussed. Panel e. The quality of the morphological data presented in this panel should be improved. The claimed delay in maturation documented by cell diameter (panel f) is not supported by the Flow profiling presented in panel d. Large cell size does not necessarily reflect delayed maturation. It may also indicate decreased cycling activity. Data on Ki67 staining should be included to prove this point.

Figure 4. Panel b. The quality of the morphological analyses needs improvement. Panel f. A WB on a pool of BFU-E and CFU-E does not add much to the information provided by Panel c. More informative would be a WB analyzing BFU-E and CFU-E separately. Panel g. the analyses of one single BFU-E and one single CFU-E does not make the point. More cells (and possibly quantitative data) are necessary to conclude that the expression of TLR8 increases between BFU-E and CFU-E. Panel i: What is the time point when the cells were analyzed? The last day of each treatment? Please clarify.

Figure 5. Lack of effects of a supernatant is not sufficient to demonstrate that the regulation of TLR8 on erythropoiesis is cell-intrinsic. Cell extrinsic mechanisms may also be exerted by cell-cell interaction. Single cell cultures is a more accepted method to assess that a mechanism is cell intrinsic. In view of the strong data on STAT5 activation presented in Figure 6, I am not sure that Figure 5 is necessary.

Figure 6. The claim that CUCPT8 synergize with EPO to improve the erythroid differentiation of CD34+ cells from cord blood should be sustained by presenting the total number of erythroid cells generated in culture. Frequency alone does not mean much. The data that stimulation of TLR8 reduces STAT5 content and activation after EPO stimulation are convincing.

Figure 7. The capture of this figure is miss-leading and should be revised to state that TLR8 inhibition improves the number of erythroid cells generated by cell lines and primary cells carrying loss of function RPS19 mutations. Also in this figure, the claim that TLR8 inhibition promotes erythropoiesis in culture should be sustained by presenting the total number of erythroid cells generated by the various experimental groups. Given the fact that the TLR8 patient, as the DBA patients, was responsive to glucocorticoids, the effects exerted by TLR8 inhibition on erythroid differentiation presented in this figure should be compared with those exerted in parallel by glucocorticoids. Panel f: Increases in STAT5 phosphorylation after CUCPT8m treatment are evident in the WB from patient 2 but not in that of patient 1. In the case of patients 1, the increase is due to the apparently reduced level of total STAT5 and is not convincing because the WB for total STAT5 is over-exposed. The conclusion should be supported by presenting WB which are not over-exposed.

Discussion. The hypothesis that inhibition of TLR8 may be an alternative strategy to treat anemia should be discussed in view of the expected immune-suppression that, as shown by the data presented in the paper currently under review, would be exerted by these drugs as side-effect. Is the immune-suppression exerted by TLR8 inhibitors different or less severe than that exerted by glucocorticoids? This part of the discussion should be better articulated.

Minor Comments

Ref 4 is miss-coded. Kim et al identified the EPO-R150Q mutation in a patient with severe anemia which was not interpreted as DBA. The sentence should be revised for precision.

Lines 57-59: "However, DBA patients fail to respond to EPO because of the impaired responsiveness or low numbers of erythroid progenitor cells^{7,13,14}". This sentence is not clear. It should be revised for precision.

Reviewer #3 (Remarks to the Author):

The manuscript by Liang et al. postulates to have identified a new gain of function(GOF) mutation of TLR8 (V434L) with association to Diamond-Blackfan anemia (DBA). Furthermore, the use of a TLR8 inhibitor improves erythropoiesis of healthy and DBA derived HSPCs.

Overall, the topic of the manuscript is innovative and interesting. However, due to missing controls, possible clonal artefacts and off-target effects of the TLR8 inhibitor and TLR8 activator, I am not yet convinced that TLR8 or a TLR8 GOF is involved in erythropoiesis. Therefore, I would recommend major revisions.

Concerns:

Page 8, line 155:

The authors should state the genome assembly: Position X:12938459 for assembly GRCh38 lies outside the ORF; Possibly the GRCh37 assembly would be correct.

Fig. 2:

Subtle differences in the NF- κ B activity of TLR8 and TLR8-V434L upon R848 or VTX2337 stimulation could be a result of differences in protein expression level. The same is true for the use of THP-1 and THP-1-TLR8-V434L (CRISPR-Cas9-mediated). I believe that the detection of TLR8 by western blotting is essential to support the authors claim. It is surprising that WT THP-1 do not respond to TLR8 ligands and contrasts results from PMID: 31778653, supplementary figure 2D (2ng/ml IL-6).

I assume that the THP-1- TLR8-V434L cells analysed are derived from a single clone and therefore it is possible that a clonal artefact is responsible for the high IL-6- induction upon TLR8 stimulation. Have the authors analysed more THP-1- TLR8-V434L clones with similar stimulatory activity? What is the origin of WT THP-1 cells? Is this a negative clone from the CRISPR/Cas9 experiment or a "heterogeneous" cell line? The argument that TLR8-V434L facilitates monocyte differentiation could also be a clonal artefact (Fig. 2d).

In summary, I am not convinced by the data presented here that the TLR8-V434L is hyperactive. Are primary monocytes from the patient and non-affected relatives available to analyse TLR8-dependent cytokine production and may support the claim of a hyperactive TLR8-V434L?

Fig. 3:

To show an effect of the TLR8-V434L variant on erythropoiesis, the authors constructed hESC carrying the corresponding SNP. Please state if male or female hESC were used.

Again, clonal artefacts could be responsible for the differential outcome. For example clone 1 and 2 behave differently in some aspects (Fig. 3c). Western blotting for TLR8 may help to substantiate the data.

Fig. 4:

Next the authors explored the potential of WT TLR8 in erythropoiesis in primary erythroid cells using an ex vivo erythroid differentiation system. The detection of TLR8 in erythroid cells and progenitors is not yet convincing.

In Fig. 4c TLR8 protein expression was detected in erythroid cells by Western blotting. Please state the TLR8-specific antibody (and isotype control for Fig 4e) used (at least I could not find it in the in the M&M section). This is essential information! Protein marker and size of the detected band are also missing (see also Fig.4f). What means “normalized to Mode” in Fig. 4e? No isotype control for Fig. 4g is depicted. The treatment of CB CD34+-derived erythroid cells with VTX2337 demonstrated a strongly reduced erythroid differentiation during 0-11 days. In contrast, the TLR8 inhibitor CU-CPT-8m had little effect. The authors should also use VTX2337 and CU-CPT-8m in combination to demonstrate that the effect of VTX2337 is really TLR8-dependent and not an off-target effect.

Fig.5:

Line 376 / 377; please correct “was resulted in by” and “TLR8 manipulation”.

In Fig. 6 the authors investigate possible mechanisms how TLR8 may influence erythropoiesis. The statement that TLR8 regulates erythropoiesis in an EPO-dependent manner seems far-fetched. Epo itself is a strong inducer of erythropoiesis and the addition of the TLR8 inhibitor CU-CPT-8m has some additional positive effect. What happens if epo and VTX2337 are combined and erythropoiesis is analysed (should be included in Figure 6a)? STAT5 phosphorylation is conversely influenced by VTX2337 and CU-CPT-8m treatment and may be involved in the effect observed. A role for ANXA2 seems possible according to the analysis in Fig. 5. I would also suggest analysing TLR8-deficient HuDEP2 cells (PMID: 31164570). If CU-CPT-8m has no effect on erythropoiesis under these conditions, then the authors claim would be substantiated.

The effect of CU-CPT-8m on adult erythropoiesis of BM-derived CD34+ HSPCs from healthy donors or RPS19-mutant DBA patient is convincing (Fig. 7). However to state that this effect is solely due to TLR8 inhibition is overstated. The use of 10 μ M CU-CPT-8m (seems a quite high concentration anyway) may cause off target effects unrelated to TLR8. The authors could strengthen their claim by using another TLR8 inhibitor such as CU-CPT9a that is even more potent and could be presumably used at lower concentrations.

RESPONSE TO CRITIQUES

We truly thank the reviewers for their instructive comments and recommendations which have improved the manuscript. We have articulated the point-by-point responses and revisions implemented below, including new experimental data and discussion that address each of the reviewers' comments and concerns. In particular, we have (1) provided additional parameters and diagnostic data of the proband to confirm our clinical diagnosis; (2) included isogenic control and the other mutant cell lines for THP1, constructed another mutant erythroid cell line (HuDEP2) for analyzing the effects of mutant TLR8 on erythropoiesis, performed rescue experiments with agonist/antagonist, included the other inhibitor and other assays to exclude off-target effects; (3) performed plasma membrane fractionation to support our proposed mechanism.

REVIEWER COMMENTS

Reviewer #1 (Remarks to the Author):

The manuscript of Liang J and colleagues reports the discovery of a homozygous gain-of-function mutation in TLR8 gene in a patient with anemia 3 weeks after birth resembling a DBA phenotype.

This study based on a case-report and first description of a mutation related to hereditary anemia aims at understanding the role of TLR8 in the control of erythropoiesis. Although number of experiments have been done using cell line models and pharmacological modulation of normal erythropoiesis with agonist and antagonist of TLR8, there are several concerns with the conclusions of experiments.

We appreciate the reviewer's positive comments and have addressed each of the concerns point by point.

Major comments

1. Regarding the case-report, some additional parameters are needed to better characterize the anemia: MCV and inflammatory parameters at diagnosis, Hb level over time and under treatment with steroids (please provide curves), red blood cell transfusion needs before and after introduction of steroids. Compared to DBA linked to heterozygous mutations in ribosomal protein genes, did the proband exhibit bone malformations? Do the authors consider the proband affected by a severe disease? Because the infant presented a homozygous mutation and the mother a heterozygous mutation, the father is expected having a heterozygous mutation: what is the Hb level of the mother and the father? Have they any symptoms?

Response:

1) Thank you for your insightful comments. We supplemented the complete blood count of the proband at diagnose: WBC $29.62 \times 10^9/L$, neutrophil count

8.96 × 10⁹/L, monocyte count 2.87 × 10⁹/L, lymphocyte count 19.58 × 10⁹/L, RBC 1.96 × 10¹²/L, hemoglobin 59 g/L, reticulocyte 0.0025 × 10¹²/L, PLT 408 × 10⁹/L, MCV 94.9 fL, MCHC 317 g/L and MCH 30.1 pg (**Figure 1b and Figure S1a**). In addition, his C-reactive protein was normal but ferritin was elevated (692.8 µg/L).

b Peripheral blood			a Peripheral blood		
Parameters	Normal range	Proband	Parameters	Normal range	Proband
WBC (×10 ⁹ /L)	4-10	29.62	Neutrophil (×10 ⁹ /L)	2-7	8.96
Hemoglobin (g/L)	100-140	59	Monocyte (×10 ⁹ /L)	0.12-1	2.87
Reticulocyte (×10 ¹² /L)	0.024-0.084	0.0025	Lymphocyte (×10 ⁹ /L)	0.8-4	19.58
PLT (×10 ⁹ /L)	100-300	408	MCHC (g/L)	320-360	317
MCV (fL)	80-100	94.9	MCH (pg)	27-34	30.1
			C-reactive protein (mg/L)	0-10	8.5

Figure 1. (b) Parameters of PB of the proband (left panel).

Figure S1. (a) Other parameters of PB of the proband (right panel).

2) He was transfusion-dependent before steroid. No further blood transfusion was required after prednisone treatment. His Hb levels over time were depicted as below (**Figure S1b**). Please find the description in the revised manuscript on page 6, lines 102-104.

Figure S1. (b) The Hb curve over time under steroids treatment and red blood cell transfusion.

3) The proband had umbilical hernia without exhibiting other physical malformations.

4) The proband had intermittent infections and was diagnosed with West

syndrome at the age of six months, but no other disease was reported. Please find the description in the revised manuscript on page 6, lines 101-102.

5) We are sorry for the confusion. As *TLR8* gene is on the X chromosome, the proband harbored the X-linked hemizygous mutation ($X^{TLR8V434LY}$). Our whole genome sequencing (WGS) data demonstrated that his father is wild type for *TLR8* gene with normal complete blood count. His mother harbored the heterozygous mutation of *TLR8* ($X^{TLR8V434LX}$). His mother (G4-17) did not manifest any discernable hematopoietic abnormalities. Please find the description in the revised manuscript on page 8, lines 147-149.

2. Regarding the West syndrome, such neurological disorders could be improved by steroids often used as first line therapy. Did the infant less affected by repeated spasms after starting the steroids? Did the authors identify one of the mutations reported in the West syndrome (ARX, STXBP1, STK9...) by whole genome sequencing?

Response: Thank you for pointing it out. After going through our WGS data, we did not find any known pathological mutations that have been reported in West syndrome¹. The detailed information is as follows:

mutant genes in West syndrome	position	mutation detection
ARX	Xp21.3	none
CDKL5	Xp22.13	none
PAFAH1B1/LIS1	17p13.3	none
DCX	Xq23	none
TUBA1A	12q13.12	none
STXBP1	9q34.11	none
KCNQ2	20q13.33	none
SPTAN	9q34.11	none
MAGI2	7q21.11	none
GRIN2A	16p13.2	none
FOXP1	14q12	none
NSD1	5q35.3	none
NEDD4	15q21.3	none
CALN1	7q11.22	none
WDR45	Xp11.23	none
SLC1A4	2p14	none

RARS2	6q15	none
UBA5	3q22.1	none
IARS2	1q41	none
PHACTR1	6p24.1	none
ATP2A2	12q24.11	none
CD99L2	Xq28	none
CLCN6	1p36.22	none
CYFIP1	15q11.2	none
CYFIP2	5q33.3	none
GNB1	1p36.33	none
GPT2	16q11.2	none
HUWE1	Xp11.22	none
KMT2A	11q23.3	none
KMT2D	12q13.12	none
MYO18A	17q11.2	none
NOS3	7q36.1	none
RYR1	19q13.2	none
RYR2	1q43	none
RYR3	15q13.3-q14	none
TAF1	Xq13.1	none
TECTA	11q23.3	none
PURA	5q31.3	none

3. TLR8 has been reported to be overexpressed in FANCC^{-/-} cells and induced inappropriate production of TNF- α (Vanderwef et al, Blood 2009). It is not clear in this manuscript whether the mutation V434L is associated or not with an increased expression level of TLR8 in THP1-mutated cells?

Response: Thank you for your helpful suggestions. We examined the TLR8 protein level in the isogenic CTRL as well as mutant THP1 cells. The results showed that TLR8 expression was not significantly affected by V434L mutation (**Figure S2c**). Please find the description in the revised manuscript on page 11, lines 228-229.

Figure S2. (c) Representative immunoblotting showing TLR8 expression in the differentiated isogenic CTRL and TLR8^{mut} THP1 cells.

4. The authors used CRISPR-cas9 strategy to edit the mutation in at least to cell models the monocytic cell line THP1 and the human ESC. They apparently used the parental cells as controls. To better establish the specificity of V434L mutation, experiments must be performed with the isogenic THP1 and hESC V434V cells must be shown.

Response: Thank you for the instructive suggestions. We have added the isogenic control (negative clone from CRISPR/cas9 screening) and one more mutant THP1 cell line and performed dosage-response assay with these cell lines accordingly (**Fig. 2d**). It showed that V434L increased the sensitivity and IL6 production after VTX2337 stimulation. Please find the description in the revised manuscript on page 11, lines 224-232.

Figure 2. (d) The IL6 production in isogenic CTRL and mutant THP1 cells responding to different dosages of VTX2337. Each THP1 cell line was induced to differentiate using PMA followed by agonist stimulation with different concentrations and IL-6 secretion analysis.

Unfortunately, we did not reserve the hESC isogenic CTRL cell lines. Since regeneration of novel ESC mutant cell lines is indeed time-consuming, we are sorry that we cannot have the isogenic CTRL cell line available and finish the related experiments during the revision. Instead, to strengthen our conclusion and to assess the function of mutant TLR8 during erythropoiesis, we constructed *TLR8^{mut}* erythroid cell lines using HuDEP2 cells, which are immortalized human erythroid progenitor cells². Consistent with results of Figure 2a-d, cell number and viability of mutant HuDEP2 cells were more

severely declined than those of isogenic CTRL after VTX2337 treatment (**Figure 2e and f**). Please find the description in the revised manuscript on page 11, lines 238-243.

Figure 2. (e) Normalized cell viability of isogenic CTRL and TLR8V434L mutant HuDEP2 cells after 10 μ M VTX2337 treatment for 24h. (f) Normalized cell numbering of isogenic CTRL and TLR8V434L mutant HuDEP2 cells after 10 μ M VTX2337 treatment for 24h.

5. Figure 2: The function of V434L mutant is assessed in monocytic THP1 cell line. The choice of this cell line is curious as the patient presented with anemia. Why do the authors use this cell line? Furthermore, they show that THP1 V434L cells in differentiation conditions upon PMA secrete IL6 at significantly higher levels than the WT THP1. This effect could be related either to the mutation suggesting an extrinsic mechanism of control of hematopoiesis, or to an off-target effect of the CRISPR-Cas9 strategy.

Response: Thank you for your constructive comments. The reason we utilized THP1 cells is mainly due to its abundant expression of TLR8 upon inducing differentiation and the well-documented signaling pathways mediated by TLR8 in these cells, which is conducive to determine TLR8 functionality. Please find the description in the revised manuscript on page 11, lines 226-227.

We totally agree with the reviewer that the enhanced production of IL6 in mutant cells implies the possible extrinsic effects of this mutation on hematopoiesis, including erythropoiesis. Consistently, a very recent study has reported that TLR8 impaired the nursing capacity of erythroblast island (EBI) macrophages

and led to fatal anemia in systemic lupus erythematosus (SLE)-prone mice³. However, in our study, we aimed at deciphering the intrinsic effect of TLR8 on erythropoiesis. We have added this part to Discussion. Please find the description in the revised manuscript on page 28, lines 690-697.

To rule out the possible off-target effects, we introduced the isogenic CTRL and one more mutant THP1 cell line. Comparing with the isogenic control, the mutant cells indeed showed higher sensitivity to the agonist treatment and IL6 production (**Figure 2d**). Please find the description in the revised manuscript on page 11, lines 228-232.

Figure 2. (d) The IL6 production in isogenic CTRL and mutant THP1 cells responding to different dosages of VTX2337. Each THP1 cell line was induced to differentiate using PMA followed by agonist stimulation with different concentrations and IL-6 secretion analysis.

6. What is the effect of the mutation in hESC on the granulomonocytic differentiation? Based on the results obtained in THP1 cells, the granulomonocytic differentiation of hESC is expected to be induced by the mutation and modulated by the agonist/antagonist of TLR8? Again, the hESC V434V cells must be used as controls.

Response: It is an interesting point. As mentioned earlier, we could not get the isogenic ESC CTRL cell lines in time. However, we repeated the colony-forming assay with larger seeding cell number (**Figure 2i and S2i**). The results showed that TLR8 mutation did not significantly affect the formation of granulomonocytic

colonies. Please find the description in the revised manuscript on page 12, lines 260-262.

Figure 2. (I, left panel) The percentage of CFUs formed by 12000 CD45⁺ HSPCs from WT and mutant ESCs treated with vehicle, 10 μ M VTX2337 or 10 μ M CUCPT8m. **Figure S2.** (I, right panel) The total number of colonies from CD45⁺ HSPCs from WT and TLR8 mutant hESCs treated with vehicle 10 μ M VTX2337 or 10 μ M CUCPT8m.

7. Fig. 3g: the effect of VTX2337 treatment on hESC V434V and hESC must be shown.

Response: Thank you for your comments. We have incorporated the effect of VTX2337 treatment on erythropoiesis in WT and mutant hESC cells (**Figure 2k and I**). The erythroid differentiation analysis and colony-forming assay showed that VTX2337 treatment further diminished erythroid differentiation with reduced CD71⁺CD235a⁺ cells and decreased generation of erythroid colonies of mutant hESC cells compared with VTX2337-treated WT cells. Please find the description in the revised manuscript on page 12, lines 258-262.

Figure 2. (k, left panel) The percentage of CD71⁺CD235a⁺ cells from WT and mutant hESCs after erythroid differentiation for 8 days with 10 μ M VTX2337. (l, right panel) The percentage of CFUs formed by 12000 CD45⁺ HSPCs from WT and mutant ESCs treated with vehicle, 10 μ M VTX2337 or 10 μ M CUCPT8m.

8. As well as the expression of TLR8 was followed along normal erythroid differentiation by western blot (Fig. 4c), it could be interesting to follow the expression of TLR8 along erythroid differentiation upon treatment with VTX2337 or CUCPT8 by flow cytometry.

Response: Thank you for your suggestion. We provided TLR8 expression during erythroid differentiation after treatment with VTX2337 or CUCPT8m at day 4, 8, 11, and 14, respectively (**Figure S3f**). The results showed that TLR8 expression was significantly higher after VTX2337 treatment while it was only slightly altered after CUCPT8m treatment. Please find the description in the revised manuscript on page 15, lines 352-354.

Figure S3. (f) TLR8 expression in erythroid cells derived from CB CD34⁺ cells at indicated time points during erythroid differentiation after VTX2337 or CUCPT8m treatment.

9. Fig. 4e-f: the authors purified BFU-E and CFU-E nicely to assess the expression of TLR8. The antibody used to visualize TLR8 recognizes an intracellular domain of the protein making cell permeabilization mandatory for flow cytometry. Additional controls in cell lines are needed to ascertain the specificity of TLR8 antibody labelling. Furthermore, the western blots to TLR8 must be shown entirely at least once in cell line (as a suppl figure).

Response: Thank you for your criticisms. We verified the specificity of the TLR8

antibody by flow cytometry in THP1 cells that expressed TLR8 (upper panel). We also assessed the specificity of TLR8 antibody used for Western blotting in HEK293T cells overexpressing TLR8 as well as THP1 cells that were treated with IFN γ to induce TLR8 expression⁴ (bottom panel, **Figure S2a**). Please find the description in the revised manuscript on page 8, lines 166-168

Representative histogram of TLR8 expression in THP1 cells. Isotype IgG was applied to confirm the specificity of TLR8 flow antibody.

Figure S2. (a) *Representative immunoblotting showing TLR8 expression in Vector/TLR8-overexpressing HEK293T cell (left panel) and THP1 cells treated with/without IFN γ (right panel).*

10. Ext Fig3a and Fig. 4: the authors stated that VTX2337 hampered the differentiation of CD34⁺-derived erythroid progenitors and precursors and induced apoptosis while CUCPT8 did the contrary. However, they referred to Ann V⁺ cells which include dead AnnV⁺ 7-AAD⁺ cells and early apoptotic AnnV⁺ 7-AAD⁻ cells. In addition, Ext Fig 3a clearly indicates that VTX 2337 at 20 microM induced cell death (not apoptosis). Altogether this suggests that the decrease of CD235a positive cells induced by VTX2337 at 10 microM is rather

due to the increase of cell death from 5 – 8% rather than a blockade of differentiation. To better address these points, the authors must show apoptotic cell percentage (AnnV+ 7-AAD-).

Response: Thank you for your critical review. For erythroid differentiation analysis by flow cytometry, we first gated the live cells with DAPI staining (DAPI+ cells) and then analyzed the proportion of CD71⁺CD235a⁺ population within the live cells. Therefore, the decreased CD235a⁺ cells induced by VTX2337 at 10 μ M was largely due to the impaired differentiation.

We also re-analyzed the apoptosis data (shown as follows). In fact, 10 μ M VTX2337 induced both apoptosis and cell death.

Early and late apoptosis of erythroid cells treated with VTX2337 or CUCPT8m at the indicated time points in erythroid cells derived from cord-blood CD34⁺ cells.

11. Fig. 5: In this interesting experiment, the authors collected conditioned media in control, VTX2337 and CUCPT8 conditions. As endpoint, the authors quantified colonies far from the contact with CM. It should be interesting to

perform multiplex cytokine/chemokine assay to address the impact of TLR8 targeting more directly.

Response: Thank you for your constructive suggestion. We performed the multiplex cytokine/chemokine assay and found that the production of the majority of cytokines regulated by TLR8 signaling in the conditioned medium was not or marginally altered (pg/ml) (**Figure S4a**). We then presumed that in our *in vitro* erythroid differentiation system, the erythroid defects could primarily arise from the intrinsic effects of TLR8 activation on erythroid cells. Please find the description in the revised manuscript on page 18, lines 424-427.

Figure S4. (a) Multiplex cytokine assay of TLR8 downstream targets in the CM, which was collected from the VTX2337 or CUCPT8m-treated inducer cells on day 4 and day 8 of erythroid differentiation, respectively.

12. Fig. 4i-w and Fig. 6: CUCPT8 did not affect cell viability, induced differentiation through an EPO-dependent mechanism implicating STAT5 localization at plasma membrane (shown by IF) due to its interaction with annexin A2. It could be interesting to perform cell fractionation to isolate plasma membrane and confirm the relocalization of pSTAT5 to the membrane with an additional method.

Response: Thank you for your instructive comment. As suggested, we carried out the plasma membrane fractionation and detected the levels of phosphorylated STAT5 (pSTAT5) and total STAT5 (tSTAT5) on the membrane with or without VTX2337 treatment. It showed that the levels of pSTAT5 and tSTAT5 on the membrane was significantly decreased after VTX2337 treatment (**Figure S4h**). Please find the description in the revised manuscript on page 23, lines 561-563.

Figure S4. (h) Representative immunoblotting showing t-STAT5 and p-STAT5 levels in the plasma membrane fraction isolated from HuDEP2 cells treated with vehicle or 10 μ M VTX2337 for 10 minutes, respectively.

Minor comments

Fig 6b: why is there two bands of GATA1 with vehicle and VTX2337 and not with CUCPT8?

Response: Thank you for the detailed review. We are deeply sorry for this confusion. We have replaced the blot in **Figure 5b**.

Figure 5. (b) Representative immunoblotting showing STAT5 phosphorylation and GATA1 expression in erythroid cells treated with 10 μ M VTX2337 or 10 μ M CUCPT8m from day 0 to 11 during erythroid differentiation of CB CD34⁺ cells.

Fig 6c-d: the time and dosage dependent decrease of pSTAT5 upon VTX2337 is not obvious. How was the cell viability after 2 hours of VTX2337?

Response: Thank you for your detailed review. We detected an apparent dosage-dependent decrease of pSTAT5 levels upon VTX2337 treatment (**Figure 5c**), while the time-dependent decrease of pSTAT5 levels were not apparent until 120 min later (**Figure 5d**). We reasoned that this might be due to the low concentration of VTX2337 (5 μ M) used.

Figure 5. (c) Representative immunoblotting (left) showing the effects of increasing VTX2337 dosage on p-STAT5 and GATA1 levels in HuDEP2 cells. The histogram (right) indicates the relative signal intensity of p-STAT5 (p-STAT5/t-STAT5). (d) Representative immunoblotting (left) showing the effects of increasing the duration of VTX2337 treatment on p-STAT5 and GATA1 levels in HuDEP2 cells. The histogram (right) indicates the relative signal intensity of p-STAT5 (p-STAT5/t-STAT5).

In addition, after supplementing with 5 μ M VTX2337 for 2h, we assessed cell viability in HuDEP2 by using Trypan Blue staining. The results showed that VTX2337 treatment under such condition did not influence the cell viability.

Cell viability of HuDEP2 cells treated with 5 μ M VTX2337 or CUCPT8m for 2h determined with Trypan Blue staining.

Ext data Fig 4c: increased p-STAT5 avec 2 hours of EPO stimulation is not expected. Rather EPO signaling turns off after 30 min upon negative

regulatory signals mediated by phosphatases and ubiquitinylation. Is it related to the HUDEP2 cells?

Response: Thank you for the helpful comments. We agreed with that the activation dynamics of EPO signaling in HuDEP2 cells might be different. Actually, results from the primary fetal liver cells, erythroid progenitor cells and pro-erythroblast cells showed that the duration of EPO signaling after activation was dynamic among different erythroid cells⁵⁻⁹.

Fig 6g: WB annex A2 of input is lacking.

Response: We included the blot of input annexin A2 in **Figure 5g**.

Figure 5. (g) Co-immunoprecipitation of ANXA2/STAT5 from HuDEP2 cells treated with 10 μ M VTX2337; left panel shows co-immunoprecipitation of ANXA2 with anti-ANXA2 antibody. Right panel shows co-immunoprecipitation of STAT5 with anti STAT5 antibody. Input levels of t-STAT5 and ANXA2 are shown. α -tubulin is used as the loading control.

Fig 6k: the control with shSCR is lacking.

Response: We included the shSCR with/without CUCPT8m challenge to **Figure 5k**.

Figure 5. (k) Representative immunoblotting showing p-STAT5 levels following 10 μ M CUCPT8m treatment in ANXA2-knockdown HuDEP2 cells.

After starvation, cells were treated with 10 μ M CUCPT8m accompanied with 3U/ml EPO for 2 h.

Reviewer #2 (Remarks to the Author):

The paper by Liang et al describes the discovery of a novel germline gain-of-function mutation in the gene encoding LTR8 in a child with anemia. These discovery lead Liang et al unveil LTR8 as a new regulator of erythropoiesis. The phenotype of the proband is strong and the genetic approach that lead to the discovery of the novel mutation is robust. The hypothesis that the mutated gene is responsible for the anemic phenotype of the proband is supported by a well-organized experimental approach which involves the analyses of the effects of the mutated gene in several surrogate models of human erythroid differentiation.

We really appreciate the reviewer's encouragement and positive comments.

I have a general clinical comment and several specific comments on the data presented in the Figures and on their interpretation.

General comment

The similarity of the anemia expressed by the proband with the macrocytic anemia which characterizes Diamond-Blackfan Anemia (DBA) is thin. The authors should provide strongest diagnostic data by moving the data provided in Table S1 in the main manuscript and by presenting photographs of the bone marrow biopsy. Alternatively, they should remove DBA from the title and mention the possibility that the proband had a novel form of DBA in the discussion. In this case, the introduction should also be revised to briefly overview all the inherited forms of anemia, including DBA. It should be noted that the paper by Kim et al (ref 4) does not mentioned that the inherited anemia induced by the EPO-R mutation is DBA, as stated by the authors. The fact that the anemia of the proband does not recapitulate DBA does not reduce the merit of the discovery of the regulatory role exerted by LTR8 in erythropoiesis reported by this paper. In this regard, it is tempting to speculate that activation of TLR8 observed during opportunistic infections and necessary

to boost the immune-response of the patients may contribute to their anemia. The paper surely include data to sustain this claim.

Response: Thank you for the insightful and constructive suggestions. As you kindly suggested, we have: 1) moved the Table S1 into the main manuscript, which shows the parameters of peripheral blood and bone marrow cell count of the proband at diagnosis (**Figure 1b and c**).

Figure 1. (b) Parameters of PB of the proband. (c) Representative Wright-Giemsa staining of bone marrow aspirates and bone marrow cell count of the proband.

2) We replaced DBA in title with inherited anemia as the proband did not exhibit the classical macrocytic anemia.

3) We incorporated a brief overview of major types of inherited anemia. Please find the description in the revised manuscript on page 3, lines 20-24.

4) We also included a short discussion describing the proband as a novel form of DBA. Please find the description in the revised manuscript on page 26-27, lines 660-661.

5) Additionally, we corrected the citation by Kim et al (ref 8). Please find the description in the revised manuscript on page 4, lines 33-34.

6) We truly agree with you that activation of TLR8 during opportunistic infections or clinical treatments boosting immune-response of the patients may contribute to the observed anemia. It is very tempting and deserves our further investigations.

Specific Comments.

Figure 1. The proband is hemizygous not homozygous as stated. The gene is on chromosome X and males have only one X and may not be homozygous. What does it mean: IV. No limit to mother? Please clarify. Please correct mother (not mather) in the Figure. The capture should clarify how the proband is indicated in the pedigree.

Response: Thank for the helpful suggestions.

1) We have revised the description as the proband should be homozygous or hemizygous if the mutation is X-linked. Please find the description in the revised manuscript on page 8, lines 150-151.

2) We are sorry for the confusion. The “No limit to mother” means we did not have any presumption whether the mother has mutation or not, as the mother manifests normal hematopoiesis without any discernable blood disease. We have replaced the “No limit to mother” as “No restriction to mother”. We also clarified it in the corresponding figure legends. Please find the description in the revised manuscript on page 7, lines 134-135.

3) The proband is indicated by the red arrow in **Figure 1a**.

Figure 1. (a) The pedigree of the proband displaying DBA-like symptoms. Grey-filled symbols indicate family members suffering from a broad range of diseases, red borders indicate family members whose PBMNCs were harvested for whole-genome sequencing (WGS), red arrow indicates the proband, and symbols with slashes represent deceased family members.

Figure 2. The rationale to investigate the activity of TLR8 on the activation of NF- κ B should be provided. Panel a. The experimental design above the bar graph should clarify what the luciferase reporter is reporting (NF- κ B activity?).

Response: Thank you for your detailed review. To unravel whether the identified mutation is functional, we utilized a well-documented signaling pathway mediated by TLR8 and NF- κ B is the canonical downstream target of TLR8. We have added the rationale in the result section. In addition, we also clarified what the luciferase reporter is reporting (NF- κ B activity) in Figure 2a. Please find the description in the revised manuscript on page 8, lines 162-163.

Figure 2. (a) Results of luciferase reporter assays carried out in HEK293T cells transfected with WT or mutant TLR8 constructs and stimulated with 5 μ M R848 or 5 μ M VTX2337. The GFP group refers to cells transfected with empty plasmids.

Panel g: are the levels of IL-6 released by the differentiated TLR8V434L cells upon stimulation significantly greater than at baseline? The differentiated cells may be releasing more IL-6 in response to VTX2337 but it is debatable that they respond to R848.

Response: Thank you for your detailed review. The statistical analysis showed that the generation of IL6 in the VTX2337-treated mutant cells upon differentiation was significantly higher than that vehicle-treated cells while R848 treatment merely led to a trend of increase. These results were in line with prior report¹⁰ that VTX2337 is a more potent agonist of TLR8 which was consistent with our results (**Figure 2a**).

(Left panel) The statistics of IL6 production by differentiated mutant THP1 cells after the stimulation of agonists.

Figure 2 (Right panel). (a) Results of luciferase reporter assays carried out in HEK293T cells transfected with WT or mutant TLR8 constructs and stimulated with 5 μ M R848 or 5 μ M VTX2337. The GFP group refers to cells transfected with empty plasmids.

Panel h: the relevance of the data presented in this panel is unclear. The structural difference between the wild-type and mutated protein is not evident. The two proteins appear structurally similar.

Response: To make it clearer, we provided the distance between two TLR8 monomers with agonist presenting (**Figure 2b**). It showed that the V>L mutation led to a closer distance between two monomers than that between wild-type counterparts which suggested that TLR8^{V434L} was easier to be activated¹¹. Please find the description in the revised manuscript on page 11, lines 222-223.

Figure 2. (b) Computational modeling of WT (V434, left panel) and mutant (p.V434L, right panel) TLR8 protein structure in the presence of agonist.

Two tailed Student's t-test is not the appropriate statistical method to analyze data sets which contain more than two groups. The data should be analyzed with a statistical method for multiple comparisons. This comment applies also to the statistical analyses of the data included in other Figures.

Response: Thank you for instructive comments. We have applied the one-way ANOVA analysis for multiple comparisons. Please find them in revised **Figure 2a**, **Figure 3o-q**, **Figure S4a-b**, and **Figure 5a-c**. Two-way ANOVA analysis was applied to **Figure S3c**.

The presentation of Panel b should be prioritized before that of panel a.

Response: We are sorry for the confusion. As you could see, panel a indicated the luciferase assay conducted in HEK293T cells while panel c (panel b in last version of the manuscript) depicted the strategy constructing TLR8 mutant THP1 cells. Our experimental logic was as following: Based on the result of luciferase reporter assay (panel a), we then investigated the structural basis of the gain-of-function mediated by the mutation (Panel b). After that, we tested the function of the mutation by constructing mutant THP1 cells panel c (**Figure 2a-d**).

Figure 2. (a) Results of luciferase reporter assays carried out in HEK293T cells transfected with WT or mutant TLR8 constructs and stimulated with 5 μM R848 or VTX2337. The GFP group refers to cells transfected with empty plasmids.

(b) Computational modeling of WT (V434, left panel) and mutant (p.V434L, right panel) TLR8 protein structure in the presence of agonist. (c) Schematic illustrating CRISPR/Cas9-mediated knock-in strategy of the mutant THP1 cells. (d) The IL6 production in isogenic CTRL and mutant THP1 cells responding to different dosages of VTX2337. Each THP1 cell line was induced to differentiate using PMA followed by agonist stimulation with different concentrations and IL-6 secretion analysis.

Take home message of Figure 2: Since un-differentiated and differentiated wild type THP1 cells do not produce IL-6 in response to the agonists, more than a gain of function, the TLR8V434L mutation appears to confer to the protein a novel function. This would be consistent with the effects of the mutations in EKLF which have been described to cause anemia by “acquisition of a new function” rather than by “gain of function”.

Response: It is an interesting hypothesis. To further certify the possible role of TLR8 mutation, we have included the isogenic CTRL THP1 cell line and the other mutant THP1 cell line followed by the treatment with different concentration of VTX2337 (**Figure 2d**). We found that *TLR8^{mut}* cells were able to respond to agonist at the lower concentration while the isogenic CTRL cells were not. These results were consistent with the structural modeling showing mutant TLR8 were easier to be activated (**Figure 2b**). Please find the description in the revised manuscript on page 11, lines 222-223 and 228-232.

Figure 2. (b, upper panel) Computational modeling of WT (V434, left panel) and mutant (p.V434L, right panel) TLR8 protein structure in the presence of agonist. (d, bottom panel) The IL6 production in isogenic CTRL and mutant THP1 cells responding to different dosages of VTX2337. Each THP1 cell line was induced to differentiate using PMA followed by agonist stimulation with different concentrations and IL-6 secretion analysis.

Figure 3: The data presented in this Figure are not robust. Panel c. The Y axes should indicate the number of cells plated. The number of cells plated was too low to allow robust assessments of the number of colonies. Differences between 4 or 1-2 BFU-E per plate are not meaningful. The experiment should be repeated plating greater number of cells. As presented now these data do not support the conclusion that TLR8V434L halts the erythroid differentiation from the BFU-E to the CFU-E stage. It is interesting that in one case, there is a significant increase in the number of CFU-GM-derived colonies. This increase is consistent with the data on the macrophage activation presented in Figure 2 and could be real. This result is also consistent with the greater WBC counts of the proband. Do DBA patients have a monocyte phenotype? To be discussed.

Response: Thank you for your insightful comments.

1) As suggested, we performed colony-forming assay with more seeding cells and consistently discovered that TLR8 mutation impaired the erythroid colony formation (**Figure 2I** and **Figure S2i**). With reducing erythroid colonies, the proportion of the myeloid colonies was increased, but not their absolute number. Please find the description in the revised manuscript on page 12, lines 260-262.

Figure 2. (I, left panel) The percentage of CFUs formed by 12000 CD45⁺ HSPCs from WT and mutant ESCs treated with vehicle, 10 μ M VTX2337 or 10 μ M CUCPT8m. **Figure S2.** (i, right panel) The total number of colonies from CD45⁺ HSPCs from WT and TLR8 mutant hESCs treated with vehicle 10 μ M VTX2337 or 10 μ M CUCPT8m.

2) The proband had a higher WBC count at diagnosis and we conjectured that could be resulted by the infections.

Panel e. The quality of the morphological data presented in this panel should be improved. The claimed delay in maturation documented by cell diameter (panel f) is not supported by the Flow profiling presented in panel d. Large cell size does not necessarily reflect delayed maturation. It may also indicate decreased cycling activity. Data on Ki67 staining should be included to prove this point.

Response: The poor quality of the morphological data presented was due to the compression of the high-resolution images. The original images were as following:

Figure 2. (i) Representative Wright-Giemsa staining of cells derived from erythroid differentiation of CD45⁺ HSPCs on day 14. Scale bar = 10 μ m.

As suggested, we analyzed the cell cycle activity with Ki67 and found that there was no significant difference in the proportion of Ki67⁺ cells between the mutant and WT cells (**Figure S2h**). Please find the description in the revised manuscript on page 12, lines 257-258.

Figure S2. (h) The percentage of Ki67⁺ cells in colonies collected on day 14, which were derived from WT and TLR8 mutant hESCs.

Figure 4. Panel b. The quality of the morphological analyses needs improvement.

Response: Thank you for your detailed review. The poor quality of the morphological data was due to the compression of the high-resolution images. The original images were as follows:

Figure 3. (b) Wright-Giemsa staining of erythroid precursors produced by ex vivo differentiation of cord blood (CB)-derived CD34⁺ HSPCs on days 4, 8, 11, and 14 of erythroid differentiation.

Panel f. A WB on a pool of BFU-E and CFU-E does not add much to the information provided by Panel c. More informative would be a WB analyzing BFU-E and CFU-E separately.

Response: Thank you for the constructive suggestion. We provided the expression of TLR8 in BFU-E and CFU-E separately by Western blotting (**Figure 3f**).

Figure 3. (f) Immunoblotting showing TLR8 expression in FACS-sorted erythroid progenitors; α -tubulin was used as the loading control.

Panel g. the analyses of one single BFU-E and one single CFU-E does not make the point. More cells (and possibly quantitative data) are necessary to conclude that the expression of TLR8 increases between BFU-E and CFU-E.

Response: Thank you for your constructive suggestion. We quantitated the fluorescence intensity of TLR8 in BFU-E and CFU-E cells (**Figure 3g**, bottom). The results showed that the signal intensity of TLR8 in BFU-E cells was higher than that in CFU-E cells

Figure 3. (g) The statistics of mean fluorescence intensity (MFI) measuring immunofluorescence of TLR8 in FACS-sorted erythroid progenitor BFU-E and CFU-E cells (EEA1, early endosome antigen 1); scale bar = 10 μ m.

Panel i: What is the time point when the cells were analyzed? The last day of each treatment? Please clarify.

Response: We are sorry for the confusion. As for the stage-specific challenge, we treated cells during the intervals of day 0-4, 4-8, 8-11, and 11-14. The erythroid differentiation and apoptosis were assessed at the last day of each treatment, namely on day 4, 8, 11, and 14, respectively. For the long-term challenge (day 0-11), cells were assessed on day 11. We have revised the schematic and figure legends for better understanding (**Figure 3h**). Please find the description in the revised manuscript on page 14, lines 304-308.

Figure 3. (h) Schematic illustrating experimental design of VTX2337 and CUCPT8m treatment during ex vivo erythroid differentiation induced in CB-derived CD34+ HSPCs. Cells were treated with VTX2337 or CUCPT8m in consecutive (days 0–11) or stage-specific manner (days 0–4, 4–8, 8–11, or 11–14) and analyzed at the indicated time points (day4, day8, day11, and day14, respectively).

Figure 5. Lack of effects of a supernatant is not sufficient to demonstrate that the regulation of TLR8 on erythropoiesis is cell-intrinsic. Cell extrinsic mechanisms may also be exerted by cell-cell interaction. Single cell cultures is a more accepted method to assess that a mechanism is cell intrinsic. In view of the strong data on STAT5 activation presented in Figure 6, I am not sure that Figure 5 is necessary.

Response: Thank you for your helpful comments. As suggested, we sorted single CFU-E cell and analyzed the erythropoiesis after the cultivation for 5 days (**Figure 4j-m**). The results showed that VTX2337 treatment decreased cell number while CUCPT8m increased cell number and the percentage of CD71+CD235a+ cells. We thus concluded that TLR8 was capable of mediating

the erythroid differentiation intrinsically. Please find the description in the revised manuscript on page 18, lines 434-438.

Figure 4. (j) Schematic illustrating experimental design. Single CFU-E cell was sorted and seeded into 96-well plates on day 5 of erythroid differentiation. These purified CFU-E cells were further induced differentiation toward erythroid lineage for 5 days with VTX32337 or CUCPT8m treatment.

(k) Cell number at day 5 of erythroid differentiation upon vehicle, VTX2337 or CUCPT8m treatment.

(l) The representative flow cytometry showing percentages of CD71⁺CD235a⁺ erythroid cells after treatment with vehicle or CUCPT8m for 5 days during erythroid differentiation.

(m) The corresponding statistical analysis of (l).

Figure 6. The claim that CUCPT8 synergize with EPO to improve the erythroid differentiation of CD34⁺ cells from cord blood should be sustained by presenting the total number of erythroid cells generated in culture. Frequency alone does not mean much. The data that stimulation of TLR8 reduces STAT5 content and activation after EPO stimulation are convincing.

Response: Thank you for the insightful comments. We provided the total cell numbers on day 4 or 8 of erythroid differentiation upon each treatment (**Figure 5a** and **S4b-c**). Consistently, we found CUCPT8m treatment enhanced the percentage of CD71⁺CD235a⁺ cells and cell numbers in EPO-dependent

manner. Please find the description in the revised manuscript on page 19, lines 447-460

Figure 5. (a) The percentage of CD71⁺CD235a⁺ cells and the total cell number of day 8 differentiated erythroid cells (originating from CB-derived CD34⁺ cells) treated with vehicle only, 3 U/ml EPO only, 10 μM CUCPT8m only, 10 μM VTX2337 only, a combination of both 3 U/ml EPO and 10 μM CUCPT8m or a combination of both 3 U/ml EPO and VTX2337 during days 4–8 of differentiation.

Figure S4. (b) The percentage of CD71⁺CD235a⁺ of and **(c)** cell number of day 8 differentiated erythroid cells (originating from CB-derived CD34⁺ cells) treated with vehicle only, 3 U/ml EPO only, 10 μM CUCPT8m only, 10 μM VTX2337 only, a combination of both 3 U/ml EPO and 10 μM CUCPT8m or a combination of both 3 U/ml EPO and 10 μM VTX2337 during days 0–8 of

differentiation.

Figure 7. The capture of this figure is miss-leading and should be revised to state that TLR8 inhibition improves the number of erythroid cells generated by cell lines and primary cells carrying loss of function RPS19 mutations. Also in this figure, the claim that TLR8 inhibition promotes erythropoiesis in culture should be sustained by presenting the total number of erythroid cells generated by the various experimental groups. Given the fact that the TLR8 patient, as the DBA patients, was responsive to glucocorticoids, the effects exerted by TLR8 inhibition on erythroid differentiation presented in this figure should be compared with those exerted in parallel by glucocorticoids.

Response: Thank you for your helpful suggestions. We have revised the caption of the figure as “TLR8 inhibition improves the number and differentiation of erythroid cells generated by cell lines and primary cells from healthy donors and DBA patients”. In addition, we calculated and incorporated the total cell number after inducing erythroid differentiation of three healthy adult BM CD34⁺ cells and two RPL5-mutant DBA CD34⁺ cells upon CUCPT8m treatment (**Figure 6h & S5b**). It showed that CUCPT8m increased cell numbers of erythroid cells in both healthy adults and DBA patients. Please find the description in the revised manuscript on page 24, lines 592-593, page 25, lines 618-620, and page 26, lines 633-634.

Figure 6. (h, left panel) Cumulative proliferation curve of BM-derived CD34⁺ cells from 2 RPL5-mutant DBA patients treated with/without TLR8 inhibitors.

Figure S5. (b, right panel) The statistics of the normalized cumulative proliferation rate.

Thank you for your perspicacious comment. In fact, in our erythroid differentiation system, glucocorticoids were supplemented during days 0-8 to ensure ample production of erythroid cells by inhibiting erythroid differentiation and thus expanding erythroid progenitor pool¹². Different from glucocorticoids, our results showed that TLR8 inhibitor promoted erythroid differentiation. Please find the description in the revised manuscript on page 29, lines 720-724.

Panel f: Increases in STAT5 phosphorylation after CUCPT8m treatment are evident in the WB from patient 2 but not in that of patient 1. In the case of patients 1, the increase is due to the apparently reduced level of total STAT5 and is not convincing because the WB for total STAT5 is over-exposed. The conclusion should be supported by presenting WB which are not over-exposed.\

Response: We are sorry for the confusion. Panel e showed the pSTAT5 and STAT5 in RPS19^{+/-} HuDEP2 cells. As total STAT5 could be the other loading control in our condition, its decrease suggested higher STAT5 phosphorylation which was consistent with our finding. To address your concern, we have replaced the previous blot with shorter exposure time (**Figure 6e**).

Figure 6. (e) Representative immunoblotting showing levels of p-STAT5 and t-STAT5 in WT and RPS19^{+/-} HuDEP2 cells treated with 3 U/ml EPO alone or in combination with 10 μM CUCPT8m for 2 hours.

Discussion. The hypothesis that inhibition of TLR8 may be an alternative strategy to treat anemia should be discussed in view of the expected immune-suppression that, as shown by the data presented in the paper currently under review, would be exerted by these drugs as side-effect. Is the immune-suppression exerted by TLR8 inhibitors different or less severe than that exerted by glucocorticoids? This part of the discussion should be better articulated.

Response: Thank you for your valuable comments. We incorporated the relevant discussion. “These findings suggest that targeting TLR8 might improve the efficacy of or restore the responsiveness to glucocorticoid. However, for possible clinical application, the dosage of TLR8 inhibitor should be carefully determined to avoid the possible immune-suppressive side-effects, especially when administrated with glucocorticoids.” Please find the description in the revised manuscript on page 29, lines 724-728.

Minor Comments

Ref 4 is miss-coded. Kim et al identified the EPO-R150Q mutation in a patient with severe anemia which was not interpreted as DBA. The sentence should be revised for precision.

Response: Thank you for your detailed review. We have revised it as “The mutation of non-RP genes including *GATA1*, *EPO*, and *ADA2* also lead to inherited anemia manifesting DBA or DBA-like phenotypes”. Please find the description in the revised manuscript on page 4, lines 33-34.

Lines 57-59: “However, DBA patients fail to respond to EPO because of the impaired responsiveness or low numbers of erythroid progenitor cells^{7,13,14}”. This sentence is not clear. It should be revised for precision.

Response: Thank you for your detailed review. We have revised this description as “DBA patients could not be treated with EPO due to either low number of the EPO-responsive progenitor cells or the low response of these

cells to EPO”¹⁴. Please find the description in the revised manuscript on page 4, lines 55-57.

Reviewer #3 (Remarks to the Author):

The manuscript by Liang et al. postulates to have identified a new gain of function (GOF) mutation of TLR8 (V434L) with association to Diamond-Blackfan anemia (DBA). Furthermore, the use of a TLR8 inhibitor improves erythropoiesis of healthy and DBA derived HSPCs.

Overall, the topic of the manuscript is innovative and interesting. However, due to missing controls, possible clonal artefacts and off-target effects of the TLR8 inhibitor and TLR8 activator, I am not yet convinced that TLR8 or a TLR8 GOF is involved in erythropoiesis. Therefore, I would recommend major revisions.

Thank you for your positive and constructive comments. We have complemented the information of isotype IgG for flow and immunofluorescence antibodies against TLR8 in the revised manuscript and confirmed the specificity of antibodies against TLR8 in different cell lines. We included the corresponding isogenic control and the other mutant THP1 cell line as well as determined TLR8 expression to avoid the possible clonal artefacts. We also have conducted the co-treatment study with antagonist and agonist, determined the effects of antagonist on TLR8 KO HuDEP2 cells as well as used the lower concentration of TLR8 inhibitors to exclude the potential off-target effects. Please find more information in the point-to-point responses.

Concerns:

Page 8, line 155:

The authors should state the genome assembly: Position X:12938459 for assembly GRCh38 lies outside the ORF; Possibly the GRCh37 assembly would be correct.

Response: Thank you for your detailed review. The mutation position was aligned to the GRCh37 assembly. Please find the description in the revised manuscript on page 8, line 155.

Fig. 2:

Subtle differences in the NF- κ B activity of TLR8 and TLR8-V434L upon R848 or VTX2337 stimulation could be a result of differences in protein expression level. The same is true for the use of THP-1 and THP-1- TLR8-V434L (CRISPR-Cas9-mediated). I believe that the detection of TLR8 by western blotting is essential to support the authors claim. It is surprising that WT THP-1 do not respond to TLR8 ligands and contrasts results from PMID: 31778653, supplementary figure 2D (2ng/ml IL-6).

Response: Thank you for the detailed review. As suggested, we determined TLR8 expression in HEK293T cell transfected with WT and mutant TLR8 plasmids after VTX2337 treatment. We found that TLR8 expression was comparable between two groups (**Figure S2b**). Please find the description in the revised manuscript on page 8-9, lines 166-168.

Figure S2. (b) Representative immunoblotting of HEK293T cells transfected with 1.5 μ g GFP/ TLR8^{WT}/TLR8^{mutant} plasmids for 24h followed by VTX2337 treatment for 24h.

Additionally, we determined the expression of TLR8 in the isogenic control (Negative clone from the CRISPR/Cas9 screening) and TLR8 mutant THP1 cells and revealed that TLR8 expression was not significantly affected (**Figure S2c**). Please find the description in the revised manuscript on page 11, lines 228-229.

Figure S2. (c) Representative immunoblotting showing TLR8 expression in the differentiated isogenic CTRL and TLR8mut THP1 cells

In the referred study, THP1 cells were induced differentiation with PMA, as described in their method section¹⁰. Besides, our results were consistent with a prior study by Natalya Odoardi et al. which showed that un-differentiated WT THP1 cells did not response to the TLR8 ligands¹⁵.

I assume that the THP-1- TLR8-V434L cells analyzed are derived from a single clone and therefore it is possible that a clonal artefact is responsible for the high IL-6- induction upon TLR8 stimulation. Have the authors analyzed more THP-1- TLR8-V434L clones with similar stimulatory activity? What is the origin of WT THP-1 cells? Is this a negative clone from the CRISPR/Cas9 experiment or a “heterogeneous” cell line? The argument that TLR8-V434L facilitates monocyte differentiation could also be a clonal artefact (Fig. 2d). In summary, I am not convinced by the data presented here that the TLR8-V434L is hyperactive.

Response: Thank you for your crucial comments. To rule out the possible clonal artefacts, we added the isogenic CTRL (Negative clone from the CRISPR/Cas9 screening) and one more mutant clone and analyzed their IL6 production upon treatment with different concentrations of TLR8 ligand (**Figure 2d**). We revealed that the two mutant THP1 cell lines were conferred with higher sensitivity and IL6 production after ligand stimulation. This result was consistent with the structural modeling showing the closer arrangement between two mutant TLR8 monomers which might reflect the more easily activated status¹⁶. We therefore concluded that the V434L mutation increased the sensitivity of TLR8 to its agonist. Please find the description in the revised manuscript on page 11, lines 222-223 and 228-232.

Figure 2. (b, upper panel) Computational modeling of WT (V434, left panel) and mutant (p.V434L, right panel) TLR8 protein structure in the presence of agonist. (d, bottom panel) The IL6 production in isogenic CTRL and mutant THP1 cells responding to different dosages of VTX2337. Each THP1 cell line was induced to differentiate using PMA followed by agonist stimulation with different concentrations and IL-6 secretion analysis.

Are primary monocytes from the patient and non-affected relatives available to analyse TLR8-dependent cytokine production and may support the claim of a hyperactive TLR8-V434L?

Response: Thank you for your insightful comments. We compared the mRNA expression of the TLR8-dependent cytokines in the peripheral blood mononuclear cells of the proband and his mother. The result showed that most downstream cytokines regulated by TLR8 were increased in the proband (**Figure S2d**). Please find the description in the revised manuscript on page 11, lines 233-235.

Figure S2. (d) Relative expression of the downstream targets of TLR8 in primary PBMCs from the proband and his mother.

Fig. 3: To show an effect of the TLR8-V434L variant on erythropoiesis, the authors constructed hESC carrying the corresponding SNP. Please state if male or female hESC were used.

Response: We confirmed the male H1 hESCs were used. Please find the description in the revised manuscript on page 31, line 791.

Again, clonal artefacts could be responsible for the differential outcome. For example clone 1 and 2 behave differently in some aspects (Fig. 3c). Western blotting for TLR8 may help to substantiate the data.

Response: We assessed the TLR8 expression in CD45⁺ hematopoietic progenitor cells derived from WT and mutant hESCs after inducing the hematopoietic differentiation. It seemed that the TLR8 expression level was comparable between two mutant cell lines (**Figure S2g**). Please find the description in the revised manuscript on page 12, lines 250-251.

Figure S2. (g) Representative immunoblotting showing TLR8 expression in WT and TLR8 mutant cells in CD45⁺ hESCs-derived HSPCs.

Fig. 4:

Next the authors explored the potential of WT TLR8 in erythropoiesis in primary erythroid cells using an ex vivo erythroid differentiation system. The detection of TLR8 in erythroid cells and progenitors is not yet convincing. In Fig. 4c TLR8 protein expression was detected in erythroid cells by Western blotting. Please state the TLR8-specific antibody (and isotype control for Fig 4e) used (at least I could not find it in the in the M&M section). This is essential information! Protein marker and size of the detected band are also missing (see also Fig.4f). What means “normalized to Mode” in Fig. 4e? No isotype control for Fig. 4g is depicted.

Response: Thank you for your constructive comments. We evaluated the TLR8 antibody specificity in HEK293T cells overexpressing TLR8 as well as in THP1 cells treated with IFN γ , which is capable of increasing TLR8 expression (**Figure S2a**). Additionally, the TLR8 antibody used for flow cytometry was tested in THP1 cells with IgG used as the control (shown as following). We also completed the isotype IgG information for **Figure 3e and 3g** in the M&M section. The antibodies used for WB, flow cytometry, and immunofluorescence were clarified in the M&M section (isotype IgG for flow antibody against TLR8 was obtained from BD Bioscience (CAT# 555749, BD Bioscience), and isotype IgG for immunofluorescence antibody against TLR8 was obtained from Sigma-Aldrich (CAT# PP64, Sigma-Aldrich). Please find the description in the revised manuscript on page 35, line 910 and page 37, line 955.

Protein size was marked in each blot throughout the revised manuscript.

Figure S2. (a, left panel) Representative immunoblotting showing TLR8 expression. Cell lysate harvested from HEK293T cells expressing TLR8 (Left) and THP1 cells treated with or without 20ng/ml IFN γ (Right).

Right panel. Representative flow cytometry showing TLR8 expression in THP1 cells. Isotype IgG was applied to isotype group.

“Normalized to Mode” was the data presentation method provided by FlowJo software. It indicated that the MFI value of each cell was normalized to the peak fluorescence intensity among cells analyzed. Then these normalized values constituted the histograms. This date-presenting way was previously used by other study¹⁷⁻¹⁹.

The treatment of CB CD34+-derived erythroid cells with VTX2337 demonstrated a strongly reduced erythroid differentiation during 0-11 days. In contrast, the TLR8 inhibitor CU-CPT-8m had little effect. The authors should also use VTX2337 and CU-CPT-8m in combination to demonstrate that the effect of VTX2337 is really TLR8-dependent and not an off-target effect.

Response: Thank you for the insightful suggestion. As suggested, we found that CUCPT8m could partially rescue the cell proliferation, differentiation, and survival impaired by VTX2337 once simultaneously supplemented into erythroid cells (**Figure 3o-q**). Please find the description in the revised manuscript on page 16, lines 366-371.

Figure 3. (o) Cell number, (p) erythroid differentiation, and (q) apoptosis of cells treated with 5 μ M VTX2337, 10 μ M CUCPT8m, or both from day 0 to day 11 of erythroid differentiation.

Fig.5: Line 376 / 377; please correct “was resulted in by” and “TLR8 manipulation”.

Response: Thank you for your detailed review. The correction has been made as “Thus, the affected erythropoiesis in VTX2337/CUCPT8m-treated cells was unlikely due to the indirect effects exerted by non-erythroid cells after TLR8 manipulation.” Please find the description in the revised manuscript on page 18, lines 427-429.

In Fig. 6 the authors investigate possible mechanisms how TLR8 may influence erythropoiesis. The statement that TLR8 regulates erythropoiesis in an EPO-dependent manner seems far-fetched. Epo itself is a strong inducer of erythropoiesis and the addition of the TLR8 inhibitor CU-CPT-8m has some additional positive effect. What happens if epo and VTX2337 are combined and erythropoiesis is analysed (should be included in Figure 6a)?

Response: We agree with the reviewer that Epo itself is a strong inducer of erythropoiesis and the addition of the TLR8 inhibitor CUCPT8m has some additional positive effects. We tuned down our hypothesis. Additionally, as suggested, we treated the erythroid cells with the combined Epo and VTX2337 (**Figure 5a** and **S4b** and **c**) and detected that VTX2337 treatment impaired the Epo-induced erythropoiesis. Thus, we linked the TLR8 with Epo signaling pathway. Please find the description in the revised manuscript on page 19, lines 447-460.

Figure 5. (a) The percentage of CD71⁺CD235a⁺ cells and the total cell number of day 8 differentiated erythroid cells (originating from CB-derived CD34⁺ cells) treated with vehicle only, 3 U/ml EPO only, 10 μM CUCPT8m only, 10 μM VTX2337 only, a combination of both 3 U/ml EPO and 10 μM CUCPT8m or a combination of both 3 U/ml EPO and VTX2337 during days 4–8 of differentiation.

Figure S4. (b) The percentage of CD71⁺CD235a⁺ of and **(c)** cell number of day 8 differentiated erythroid cells (originating from CB-derived CD34⁺ cells) treated with vehicle only, 3 U/ml EPO only, 10 μM CUCPT8m only, 10 μM VTX2337 only, a combination of both 3 U/ml EPO and 10 μM CUCPT8m or a combination of both 3 U/ml EPO and 10 μM VTX2337 during days 0–8 of differentiation.

STAT5 phosphorylation is conversely influenced by VTX2337 and CU-CPT-8m treatment and may be involved in the effect observed. A role for ANXA2 seems possible according to the analysis in Fig. 5. I would also suggest analyzing TLR8-deficient HuDEP2 cells (PMID: 31164570). If CU-CPT-8m has no effect on erythropoiesis under these conditions, then the authors claim would be substantiated.

Response: It is a good idea. We constructed the TLR8 KO HuDEP2 cell lines by CRISPR/Cas technology and analyzed the effects of CUCPT8m on these cells in regard to STAT5 phosphorylation and cell viability (**Figure 5l** and **m**). The results indicated that CUCPT8m treatment increased the pSTAT5 level and cell viability of isogenic CTRL cells that was abolished in TLR8 KO cells,

underscoring effects of CUCPT8m were largely dependent on TLR8. Please find the description in the revised manuscript on page 23, lines 572-576.

Figure 5. (I) Representative immunoblotting showing p-STAT5 levels following 5 μ M CUCPT8m treatment in TLR8 knockout HuDEP2 cells. After starvation, cells are treated with 5 μ M CUCPT8m accompanied with 3U/ml EPO for 2 h. (m) Normalized cell viability of TLR8 knockout HuDEP2 cells after 5 μ M CUCPT8m treatment for 24 h.

The effect of CU-CPT-8m on adult erythropoiesis of BM-derived CD34+ HSPCs from healthy donors or RPS19-mutant DBA patient is convincing (Fig. 7). However, to state that this effect is solely due to TLR8 inhibition is overstated. The use of 10 μ M CU-CPT-8m (seems a quite high concentration anyway) may cause off target effects unrelated to TLR8. The authors could strengthen their claim by using another TLR8 inhibitor such as CU-CPT9a that is even more potent and could be presumably used at lower concentrations.

Response: We supplemented the treatment with lower concentration of CUCPT8m (5 μ M) and another TLR8 inhibitor CUCUP9a, as suggested by the reviewer (**Figure 6h** and **S5a-b**). Consistently, CUCPT8m at 5 μ M increased cell number of erythroid cells derived from BM of healthy adults and two *RPL5*-mutant DBA patients. Whereas, 1 μ M CUCPT9a only exhibited a trend of increasing cell number in erythroid cells derived from BM of two *RPL5*-mutant DBA patients. One of the speculations of such marginal effect of CUCPT9a could be the lower concentration used. However, we did not perform further tests due to rare DBA patient samples available and the limited revision window. We will investigate the effects of CUCPT9a on erythropoiesis under healthy and

pathological condition in the future studies. Please find the description in the revised manuscript on page 25, lines 618-620 and page 26, lines 633-634.

Figure 6. (h) Cumulative proliferation curve of BM-derived CD34⁺ cells from 2 RPL5-mutant DBA patients treated with/without TLR8 inhibitors.

Figure S5. (a) Representative flow cytometry showing erythroid differentiation of cells from healthy adult BM on day 11 of differentiation and the statistics of the relative MFI of CD235a. **(b)** The statistics of the normalized cumulative proliferation rate.

REFERENCES

1. Pavone, P. et al. West syndrome: a comprehensive review. *Neurological Sciences* 41, 3547-3562 (2020).
2. Kurita, R. et al. Establishment of Immortalized Human Erythroid Progenitor Cell Lines Able to Produce Enucleated Red Blood Cells. *PLOS ONE* 8, e59890 (2013).
3. Naomi, I.M. et al. Human TLR8 induces inflammatory bone marrow erythromyeloblastic islands and anemia in SLE-prone mice. *Life Science Alliance* 6, e202302241 (2023).
4. Ishii, N., Funami, K., Tatematsu, M., Seya, T. & Matsumoto, M. Endosomal Localization of TLR8 Confers Distinctive Proteolytic Processing on Human Myeloid Cells. *The Journal of Immunology* 193, 5118-5128 (2014).
5. Krüger, A. et al. Human TLR8 senses UR/URR motifs in bacterial and mitochondrial RNA. *EMBO reports* 16, 1656-1663 (2015).

6. Ostendorf, T. et al. Immune Sensing of Synthetic, Bacterial, and Protozoan RNA by Toll-like Receptor 8 Requires Coordinated Processing by RNase T2 and RNase 2. *Immunity* 52, 591-605.e596 (2020).
7. Fan, Y. et al. Deletion of a flippase subunit Tmem30a in hematopoietic cells impairs mouse fetal liver erythropoiesis. *Haematologica* 104, 1984-1994 (2019).
8. Tsuji-Takayama, K. et al. Erythropoietin induces sustained phosphorylation of STAT5 in primitive but not definitive erythrocytes generated from mouse embryonic stem cells. *Experimental Hematology* 34, 1323-1332 (2006).
9. Penta, K. & Sawyer, S.T. Erythropoietin Induces the Tyrosine Phosphorylation, Nuclear Translocation, and DNA Binding of STAT1 and STAT5 in Erythroid Cells (*). *Journal of Biological Chemistry* 270, 31282-31287 (1995).
10. Greulich, W. et al. TLR8 Is a Sensor of RNase T2 Degradation Products. *Cell* 179, 1264-1275.e1213 (2019).
11. Tanji, H., Ohto, U., Shibata, T., Miyake, K. & Shimizu, T. Structural Reorganization of the Toll-Like Receptor 8 Dimer Induced by Agonistic Ligands. *Science* 339, 1426-1429 (2013).
12. Ashley, R.J. et al. Steroid resistance in Diamond Blackfan anemia associates with p57Kip2 dysregulation in erythroid progenitors. *The Journal of Clinical Investigation* 130, 2097-2110 (2020).
13. Taves, M.D. et al. Tumors produce glucocorticoids by metabolite recycling, not synthesis, and activate Tregs to promote growth. *The Journal of Clinical Investigation* 133 (2023).
14. Lee, H.-Y. et al. PPAR- α and glucocorticoid receptor synergize to promote erythroid progenitor self-renewal. *Nature* 522, 474-477 (2015).
15. Odoardi, N., Kourko, O., Petes, C., Basta, S. & Gee, K. TLR7 Ligation Inhibits TLR8 Responsiveness in IL-27-Primed Human THP-1 Monocytes and Macrophages. *Journal of Innate Immunity* 13, 345-358 (2021).
16. Zhang, S. et al. Small-molecule inhibition of TLR8 through stabilization of its resting state. *Nature Chemical Biology* 14, 58-64 (2018).
17. Cai, X.T. et al. Gut cytokines modulate olfaction through metabolic reprogramming of glia. *Nature* 596, 97-102 (2021).
18. Tanimura, N. et al. GATA/Heme Multi-omics Reveals a Trace Metal-Dependent Cellular Differentiation Mechanism. *Developmental Cell* 46, 581-594.e584 (2018).
19. Soukup, A.A.-O., Matson, D.A.-O., Liu, P., Johnson, K.A.-O. & Bresnick, E.A.-O. Conditionally pathogenic genetic variants of a hematopoietic disease-suppressing enhancer. *Sci Adv.* 7, 2375-2548 (Electronic) (2021).

REVIEWER COMMENTS

Reviewer #1 (Remarks to the Author):

The authors added numerous informations to revise their manuscript.

Comments 1 and 2: Clinical data of the proband and his family have been completed. Known mutations reported in the West syndrome have been excluded.

Comment 3: The expression level of TLR8 in THP1 isogenic and mutant clones was been addressed by Western blot. The quality is not as good as expected since the tubulin loading control is overexposed and therefore not quantifiable. As this experiment is mandatory to understand the functional consequence of the mutation, I suggest to improve the quality in order to provide a quantification of the Western blot. Since the authors kindly provided a flow cytometry histogram showing the TLR8 detection in THP1 wild-type cells using a specific antibody, the quantification of TLR8 expression level by flow cytometry is also possible and will be useful.

Comment 4: The authors have provided experiments using an isogenic TLR8 wild-type clone as requested together with an additional CRISPR-Cas9 mutant clone. However, they did not obtain the isogenic ESC and provided instead isogenic and mutant HUDEP2 clones, which is acceptable to me. Histograms showing the normalized cell viability and cell number of these HUDEP2 clones upon TLR8 agonist VTX2337 treatment are provided in Fig2e and 2f. "Normalized" is supposed to indicate a ratio of treated to untreated cells. This should be clarified in the legend.

Comments 5-7: In Figure 2i, it is more pertinent to show the total numbers of colonies (currently in extended figure 2i) rather than the percentages of colonies.

It is still unclear whether the TLR8 mutation is blocking the erythroid differentiation of hESC or inducing cell death. Only a double labelling of erythroid subpopulations with erythroid markers and annexin V together with absolute quantification of erythroid subpopulations along the differentiation process could provide this information.

The TLR8 agonist treatment of hESC may have killed the progenitors in colony assays as a decreased number of colonies rather than decrease of colony size is reported. At the precursor level, the flow cytometry experiment shown in fig 2k suggested a reduction of mature erythroblasts that is very likely due to increased sensitivity of the cells to death as shown in the figure provided in response to my comment 10.

It is still difficult to assess whether changes in TLR8 expression along erythroid differentiation may cause an increased sensitivity to cell death in the absence of TLR8 agonist because the authors provided separate Western blots of the different days of culture instead of a unique blot.

Comments 11 & 12 and minor points: I thank the authors to clarify these points.

Reviewer #2 (Remarks to the Author):

The paper by Dr. Shi et al reports that the phenotype of a child carrying a mutation in the TLR8 gene includes high WBC counts and anemia. The anemia was treatable by steroids. The paper then presents loss and gain of function experiments in human primary cells, in cells from patients with Diamond Blackfan Anemia (DBA) and in erythroid cell lines indicating that TLR8 sustains an annexin A2-mediated localization of STAT5 to the plasma membrane that modulates signaling from the erythropoietin receptor.

Although I am pleased that in their rebuttal letter the authors recognized that the diagnosis of this patient as DBA is an overstatement, the abstract, introduction and discussion have not been sufficiently revised and they are still focused on DBA. This is imprecise, potentially irrelevant and confusing for the field.

There are additional aspects of the revised manuscript that needs improvement:

1) In response to my request, the authors have added the photo of the bone marrow aspirate (Figure 1C). The field presented is too small and the photo is not representative of a bone marrow aspirate. It seems that the aspirate failed, and that the sample is mostly peripheral blood. This is not uncommon when sampling bone marrow. The cells pictured in the photos are mostly neutrophils which suggest that the increases in WBC observed in the blood are due to neutrophils, still the authors present data claiming that the mutation increases monocyte proliferation. Differential blood counts are not presented to sustain this claim. This is confusing.

2) The reply to the question by reviewer 1 on Fig. 6b about GATA1 being two bands in vehicle and VTX2337 and being one band in cells with CUCPT8 is unclear. The replacement of the WB presented in the Figure with another from a different experiment showing one band in all the samples is confusing. How many times the biochemical experiments have been repeated and how representative is the WB presented in the new figure of the entire set of data? This needs clarification.

3) The link between modulation of TLR8 activity and erythropoietin signaling presented in Figure 4 is thin. It is based on WB determinations of the localization of STAT5 on the plasma membrane and on the levels of pSTAT5 phosphorylation in cells exposed to 3 units erythropoietin at a single time point. Three units of erythropoietin are super-physiologic concentrations. The conclusion should be supported by data obtained with control cells analysed in parallel without erythropoietin and exposed to increasing erythropoietin concentration. Ideally these experiments should also include a time course since the levels of STAT5 phosphorylation in cells exposed to erythropoietin are known to increase at specific time points after exposure. Without the level of STAT5 phosphorylation in the absence of erythropoietin, in cells exposed to low erythropoietin levels for increasing period of time, no conclusion can be made on the effects of TLR8 and erythropoietin signaling.

4) In addition to DBA, treatment with steroids increases hemoglobin levels and makes transfusion independent a variety of conditions, including some subtypes of myelodysplastic syndrome and of myeloproliferative disorders. Steroids may even induce polycythemia in allergic individuals with normal erythropoietin response. The data on the erythropoietin signaling presented provide limited insights on the mechanisms that made the TLR8 patient transfusion independent.

Minor:

- My comment on the need to improve the morphological data is not addressed. The photos of the morphological data still do not include scale bars, the quality of Figure 3b at greater resolution provided by the authors is still low, Figure 3g should present more cells. The photo of only one cell per type is not convincing.
- Figure 4: the levels of STAT5 phosphorylation should be presented as stoichiometric values with respect to total STAT5 levels.

Reviewer #3 (Remarks to the Author):

The authors addressed most of my concerns convincingly.

For Figure 5l, please adjust upper labelling, short lines are mislocated. The authors should include WB data showing that HuDEP2 cells express TLR8 and the corresponding TLR8 ko cells don't.

In summary, the manuscript improved significantly and I recommend publication after addressing my final concern.

Response to Critiques

Reviewer #1 (Remarks to the Author):

The authors added numerous informations to revise their manuscript.

Comments 1 and 2: Clinical data of the proband and his family have been completed. Known mutations reported in the West syndrome have been excluded.

Response: We appreciate your valuable feedback on improving the quality of our manuscript.

Comment 3: The expression level of TLR8 in THP1 isogenic and mutant clones was been addressed by Western blot. The quality is not as good as expected since the tubulin loading control is overexposed and therefore not quantifiable. As this experiment is mandatory to understand the functional consequence of the mutation, I suggest to improve the quality in order to provide a quantification of the Western blot. Since the authors kindly provided a flow cytometry histogram showing the TLR8 detection in THP1 wild-type cells using a specific antibody, the quantification of TLR8 expression level by flow cytometry is also possible and will be useful.

Response: Thank you for your critical and insightful comments. We have re-conducted the Western blot and quantitated the relative expression of TLR8 in the isogenic and mutant THP1 cells from three biological replicates (**Extended Data Fig. 2c**). It showed that TLR8 expression was comparable among the isogenic and mutant cells. Please find the related text on page 11 and line 219-221.

Extended Data Fig. 2. (c) The representative Western blot of TLR8 expression in the isogenic CTRL and mutant THP1 cells (left panel) as well as the corresponding quantitation (right panel).

Flow cytometry was also applied to determine TLR8 expression. We included the quantification of TLR8 MFI of each group in the revised manuscript (**Extended Data Fig. 2d**). It turned out that there was no significant difference of TLR8 expression among the isogenic and mutant THP1 cells, consistent with WB assays. Please find the related text

on page 11 and line 219-221.

Extended Data Fig. 2 (d) The representative flow cytometry (left panel) of TLR8 expression in the isogenic CTRL and mutant THP1 cells as well as the quantitation of MFI of TLR8 in each group (right panel).

Comment 4: The authors have provided experiments using an isogenic TLR8 wild-type clone as requested together with an additional CRISPR-Cas9 mutant clone. However, they did not obtain the isogenic ESC and provided instead isogenic and mutant HUDEP2 clones, which is acceptable to me. Histograms showing the normalized cell viability and cell number of these HUDEP2 clones upon TLR8 agonist VTX2337 treatment are provided in Fig2e and 2f. “Normalized” is supposed to indicate a ratio of treated to untreated cells. This should be clarified in the legend.

Response: We apologize for the lack of clarity. As you indicated, “Normalized” indeed refers to the ratio of treated to untreated cells. We have included this information to the revised figure legend of **Fig.2e and 2f**. Please find the related text on page 10 and line 195-198.

Comments 5-7: In Figure 2i, it is more pertinent to show the total numbers of colonies (currently in extended data figure 2i) rather than the percentages of colonies.

Response: It is helpful. We have replaced the panel of the total numbers of colonies to **Fig 2I** and the percentages of colonies to the **Extended Data Fig. 2I**.

It is still unclear whether the TLR8 mutation is blocking the erythroid differentiation of hESC or inducing cell death. Only a double labelling of erythroid subpopulations with erythroid markers and annexin V together with absolute quantification of erythroid subpopulations along the differentiation process could provide this information.

Response: Thanks for pointing out the considerable issue and suggestions. We have labeled the WT and TLR8 mutant cells with anti- CD71/CD235a/Annexin V-FITC antibodies/protein at day 8 of erythroid differentiation derived from hESCs for flow cytometry analysis. The results indicated that their apoptosis was not significantly different which were incorporated in **Extended Data Fig. 2j**. However, the cell number of

CD71⁺CD235a⁺ erythroblasts in mutant cells was fewer than in WT ones. It thus suggested that apoptosis did not contribute greatly to the reduced generation of CD71⁺CD235a⁺ cells, which was possibly induced by impaired erythroid differentiation. Please find the related text on page 12 and line 251-255.

Flow cytometry analysis of Annexin V⁺ cells in CD71⁺CD235a⁺ erythroid subpopulation (left panel) and cell number of CD71⁺CD235a⁺ erythroid subpopulation (right panel) of WT and mutant hESCs at day 8 of erythroid differentiation.

The TLR8 agonist treatment of hESC may have killed the progenitors in colony assays as a decreased number of colonies rather than decrease of colony size is reported.

Response: Thank you for your critical comments. As TLR8 agonist treatment decreases CFU-E in CD34⁺ cells as showed in **Fig. 3n**, we analyzed the effect of TLR8 agonist upon the apoptosis of CFU-E. Apoptosis analysis indicated that TLR8 agonist increased CFU-E apoptosis (**Extended Data Fig. 3i**). Therefore, the decreased number of colonies was at least partially contributed by the increased apoptosis of the progenitors. Please find the related text on page 16 and line 376-377.

Fig 3. (n) Colony-forming assay of CD34⁺ cells treated with agonist and antagonist (right panels) and **Extended Data Fig 3. (i)** Apoptosis analysis of CFU-E by flow cytometry at day6 of erythroid differentiation of CD34⁺ cells (right panel).

At the precursor level, the flow cytometry experiment shown in fig 2k suggested a reduction of mature erythroblasts that is very likely due to increased sensitivity of the cells to death as shown in the figure provided in response to my comment 10.

Response: Thank you for your critical and insightful comments. We analyzed the cell apoptosis of CD71⁺CD235a⁺ cells and found that TLR8 agonist showed little effect upon their apoptosis (*Extended Data Fig. 2j*). Please find the related text on page 12 and line 251-255.

Extended Data Fig 2. (j) Flow cytometry analysis of Annexin V⁺ cells in the erythroid subpopulation of WT and mutant hESCs with/without VTX2337 treatment at day 8 of erythroid differentiation.

It is still difficult to assess whether changes in TLR8 expression along erythroid differentiation may cause an increased sensitivity to cell death in the absence of TLR8 agonist because the authors provided separate Western blots of the different days of culture instead of a unique blot.

Response: We apologize for this inconvenience. After assaying TLR8 expression during erythroid differentiation with a unique blot, we found that TLR8 expression decreased during erythroid maturation (*Extended Data Fig. 3h*). However, it increased when cells were treated with agonist. Please find the related text on page 16 and line 359-362.

Extended Data Fig3. (h) TLR8 expression in erythroid cells derived from CB-derived CD34⁺ cells at indicated time points during erythroid differentiation after 10 μ M VTX2337 or 10 μ M CUCPT8m treatment. GAPDH is used as the loading control (n = 3).

Comments 11 & 12 and minor points: I thank the authors to clarify these points.

Response: We appreciate your valuable feedback on improving the quality of our manuscript.

Reviewer #2 (Remarks to the Author):

The paper by Dr. Shi et al reports that the phenotype of a child carrying a mutation in the TLR8 gene includes high WBC counts and anemia. The anemia was treatable by steroids. The paper then presents loss and gain of function experiments in human primary cells, in cells from patients with Diamond Blackfan Anemia (DBA) and in erythroid cell lines indicating that TLR8 sustains an annexin A2-mediated localization of STAT5 to the plasma membrane that modulates signaling from the erythropoietin receptor.

Although I am pleased that in their rebuttal letter the authors recognized that the diagnosis of this patient as DBA is an overstatement, the abstract, introduction and discussion have not been sufficiently revised and they are still focused on DBA. This is imprecise, potentially irrelevant and confusing for the field.

Response: We appreciate your valuable feedback on improving the quality of our manuscript and recognize that additional improvement are needed. We have revised this confusing diagnosis and also sufficiently revised the related information in the abstract, introduction and discussion pertinent to DBA. Please find them in the abstract (lines 2-6, page 3), introduction (lines 20-58, page 3-5) and discussion (lines 695-715, page 28-29).

There are additional aspects of the revised manuscript that needs improvement:

1) In response to my request, the authors have added the photo of the bone marrow aspirate (Figure 1C). The field presented is too small and the photo is not representative of a bone marrow aspirate. It seems that the aspirate failed, and that the sample is mostly peripheral blood. This is not uncommon when sampling bone marrow. The cells pictured in the photos are mostly neutrophils which suggest that the increases in WBC observed in the blood are due to neutrophils, still the authors present data claiming that the mutation increases monocyte proliferation. Differential blood counts are not presented to sustain this claim. This is confusing.

Response: Thank you for your critical comments. We are deeply sorry for the confusion. We have provided more photos (1000×) acquired from the same bone marrow smear of the proband as following and replaced the photo with another representative one in **Figure 1c**.

In the proband's bone marrow, 78% karyocyte cells were granulocytes (normal range 40-60%), 4.5% monocytic series (normal range about 0-3%) while no erythroid cell (normal range about 15-25%) were found (The information has been incorporated in **Figure 1c**). The counts of granulocytes and monocytic series were all higher than those in the normal range. These results were consistent with the counts of peripheral blood with increased

neutrophils ($8.95 \times 10^9/L$, normal range $2 \sim 7 \times 10^9/L$) and monocyte ($2.87 \times 10^9/L$, normal range $0.12 \sim 1 \times 10^9/L$).

2) The replay to the question by reviewer 1 on Fig. 6b about GATA1 being two bands in vehicle and VTX2337 and being one band in cells with CUCPT8 is unclear. The replacement of the WB presented in the Figure with another from a different experiment showing one band in all the samples is confusing. How many times the biochemical experiments have been repeated and how representative is the WB presented in the new figure of the entire set of data? This needs clarification.

Response: We are deeply sorry for this confusion. The reason for “two bands” of GATA1 in vehicle and VTX2337 while one band in CUCPT8m was due to our unintended cropping of the membrane during the experiment (highlighted by green box, left panel). Actually, GATA1 shows one band in each group. The biological triplicate were provided in the right panel.

The original blots with unintended cropping of GATA1 into two bands (left panel). The right panel showing the biological triplicate of the expression of p-STAT5, total STAT5 and GATA1 in CD34⁺ cells treated with agonist/antagonists.

We totally agree with you that all the blots in one Western blot must be obtained from the same samples treated under the same condition. Actually, in last revised manuscript, we replaced not only the band of GATA1 but also other bands obtained under the same condition.

Additionally, we have provided the original blots of biological triplicates of each WB result presented in the manuscript in the **Supplementary Information file** during this revision.

3) The link between modulation of LTR8 activity and erythropoietin signaling presented in Figure 4 is thin. It is based on WB determinations of the localization of STAT5 on the plasma membrane and on the levels of pSTAT5 phosphorylation in cells exposed to 3 units erythropoietin at a single time point. Three units of erythropoietin are super-physiologic concentrations. The conclusion should be supported by data obtained with control cells analysed in parallel without erythropoietin and exposed to increasing erythropoietin concentration. Ideally these experiments should also include a time course since the levels of STAT5 phosphorylation in cells exposed to erythropoietin are known to increase at specific time points after exposure. Without the level of STAT5 phosphorylation in the absence of erythropoietin, in cells exposed to low erythropoietin levels for increasing period of time, no conclusion can be made on the effects of LTR8 and erythropoietin signaling.

Response: Thank you for your valuable comments. We determined the time-dependency of STAT5 phosphorylation at lower EPO concentration (0.5 UI/ml) as well as the effect of TLR8 activation on STAT5 phosphorylation upon various EPO dosages (0 UI/ml, 0.5 UI/ml, and 3 UI/ml) (**Extended Data Fig. 4d and 4e**). The results showed that EPO significantly increased STAT5 phosphorylation in a time-dependent manner (**Extended Data Fig. 4d**). More importantly, such concentration-dependent increase of STAT5 phosphorylation induced by EPO was significantly diminished by TLR8 activation (**Extended Data Fig. 4e**). We thus linked TLR8 activity and EPO signaling and concluded TLR8 activation could interfere EPO signaling. Please find the related text on page 20 and line 478-482.

Extended Data Figure 4. (d) Representative immunoblotting showing STAT5 phosphorylation in HuDEP2 cells treated with 0.5 U/ml EPO for 30, 60, or 120 minutes and the corresponding quantitation. α-tubulin is used as the loading control. (e) Representative

immunoblotting showing STAT5 phosphorylation in HuDEP2 cells treated with 0, 0.5, or 3 U/ml EPO in combination with 10 μ M VTX2337 for 120 minutes and the quantitation.

4) In addition to DBA, treatment with steroids increases hemoglobin levels and makes transfusion independent a variety of conditions, including some subtypes of myelodysplastic syndrome and of myeloproliferative disorders. Steroids may even induce polycythemia in allergic individuals with normal erythropoietin response. The data on the erythropoietin signaling presented provide limited insights on the mechanisms that made the LTR8 patient transfusion independent.

Response: Thank you for pointing out the considerable issue. We agree with you that steroids play pivotal roles in anemia treatment independent of transfusion and even induce polycythemia under some conditions¹⁻³. Mechanistically, although steroids improve erythropoiesis by several EPO-independent way, it has been also reported that steroids can improve erythropoiesis through an EPO-dependent way^{4,5}. In this study, the proband carrying the GOF TLR8 mutation became transfusion-independent after glucocorticoid treatment. Prior study has reported that glucocorticoid could restrain the activity of TLR7 and TLR8 in dendritic cells⁶. By linking the glucocorticoid with TLR8, we then speculate that the therapeutic efficacy of the proband might associate with the reduced TLR8 activity and the resultant activation of EPO signaling pathway via JAK-STAT in erythroid cells. Please find the related text on page 28-29 and line 709-715.

Minor:

- My comment on the need to improve the morphological data is not addressed. The photos of the morphological data still do not include scale bars, the quality of Figure 3b at greater resolution provided by the authors is still low, Figure 3g should present more cells. The photo of only one cell per type is not convincing.

Response: We apologize for our unsatisfied response and appreciate the time and effort you put into our manuscript. We have provided the new photos with more cells and scale bar in the presented manuscript. The high-quality morphological data were shown as follows with scale bars (**Fig 2i**). Meanwhile, we provided new photos for **Fig 3b** and **3g** with more cells.

Figure 2. (i) Wright-Giemsa staining of cells derived from erythroid differentiation of CD45⁺

HSPCs on day 14; scale bar = 10 μm .

Figure 3. (b) Wright-Giemsa staining of cells produced by *ex vivo* differentiation of cord blood-derived CD34⁺ HSPCs on days 4, 8, 11, and 14 of erythroid differentiation; scale bar = 10 μm .

Figure 3. (g) Representative immunofluorescence of TLR8 in FACS-sorted erythroid

progenitor BFU-E and CFU-E cells and the quantitation of fluorescence intensity (bottom panel); scale bar = 10 μ m.

- Figure 4: the levels of STAT5 phosphorylation should be presented as stoichiometric values with respect to total STAT5 levels.

Response: Thank you for your valuable comments. We have added this information in the Fig 5b-e, 5j-l, Extended Data Fig. 4d, 4e, 4g, and Fig 6e.

Reviewer #3 (Remarks to the Author):

The authors addressed most of my concerns convincingly.

Response: Thank you for your encouragement.

For Figure 5l, please adjust upper labelling, short lines are mislocated. The authors should include WB data showing that HuDEP2 cells express TLR8 and the corresponding TLR8 ko cells don't.

Response: Thank you for your detailed review and valuable suggestion. We have adjusted the labelling and included the expression of TLR8 in isogenic and TLR8 knockout HuDEP2 cells by WB (*Extended Data Fig. 4m*). Please find the related text on page 24 and line 591-592.

Extended Data Figure 4. (m) Representative immunoblotting (left panel) showing TLR8 expression in isogenic CTRL and TLR8-knockout HuDEP2 cells and the quantitation (right panel).

In summary, the manuscript improved significantly and I recommend publication after addressing my final concern.

We thank you for the assessment and suggestions and would like to express our gratitude for your positive agreement to accept the current manuscript.

References

- 1 Nina, L., Satnaam, B., Don, P. & Hossein, A. Gerstmann syndrome complicating polycythemia secondary to anabolic steroid use. *BMJ Case Reports* **12**, e229004, doi:10.1136/bcr-2018-229004 (2019).
- 2 Low, M. S. Y., Vilcassim, S., Fedele, P. & Grigoriadis, G. Anabolic androgenic steroids, an easily forgotten cause of polycythaemia and cerebral infarction. *Internal Medicine Journal* **46**, 497-499, doi:<https://doi.org/10.1111/imj.13029> (2016).
- 3 Ballin, A. *et al.* Steroid therapy may be effective in augmenting hemoglobin levels during hemolytic crises in children with hereditary spherocytosis. *Pediatric Blood & Cancer* **57**, 303-305, doi:<https://doi.org/10.1002/pbc.22844> (2011).
- 4 Lee, H.-Y. *et al.* PPAR- α and glucocorticoid receptor synergize to promote erythroid progenitor self-renewal. *Nature* **522**, 474-477, doi:10.1038/nature14326 (2015).
- 5 Ohene-Abuakwa, Y., Orfali, K. A., Marius, C. & Ball, S. E. Two-phase culture in Diamond Blackfan anemia: localization of erythroid defect. *Blood* **105**, 838-846, doi:<https://doi.org/10.1182/blood-2004-03-1016> (2005).
- 6 Larangé, A., Antonios, D., Pallardy, M. & Kerdine-Römer, S. Glucocorticoids inhibit dendritic cell maturation induced by Toll-like receptor 7 and Toll-like receptor 8. *Journal of Leukocyte Biology* **91**, 105-117, doi:10.1189/jlb.1110615 (2011).

REVIEWERS' COMMENTS

Reviewer #1 (Remarks to the Author):

The authors have provided all requested experiments. I have no additional comments.

Reviewer #2 (Remarks to the Author):

All my comments are addressed. Thanks

Reviewer #3 (Remarks to the Author):

The authors have addressed all my latests concerns adequately.